# Temporally resolved and interpretable machine learning model of GPCR conformational transition

Babgen Manookian [1,4], Elizaveta Mukhaleva[1,2,4], Grigoriy Gogoshin [1], Supriyo Bhattacharya [1], Sivaraj Sivaramakrishnan [3], Nagarajan Vaidehi [1,2] ✉, Andrei S. Rodin [1,2] ✉ & Sergio Branciamore [1,2] ✉

Identifying target-specific drugs remains a challenge in pharmacology, especially for highly homologous proteins such as dopamine receptors $D_2R$ and $D_3R$. Differences in target-specific cryptic druggable sites for such receptors arise from the distinct conformational ensembles underlying their dynamic behavior. While Molecular Dynamics (MD) simulations has emerged as a powerful tool for dissecting protein dynamics, the sheer volume of MD data requires scalable and unbiased data analysis strategies to pinpoint residue communities regulating conformational state ensembles. We present the Dynamically Resolved Universal Model for BayEsiAn network Tracking (DRUMBEAT) interpretable machine learning algorithm and validate it by identifying residue communities that enable the deactivation of the $\beta_2$-adrenergic receptor. Further, upon analyzing dopamine receptor dynamics we identify distinct and non-conserved residue communities around the contacts $F170^{4.62}\_F172^{ECL2}$ and $S146^{4.38}\_G141^{34.56}$ that are specific to $D_3R$ conformational transitions compared to $D_2R$. This information can be tapped to design subtype-specific drugs for neuropsychiatric and substance use disorders.

G-protein-coupled receptors (GPCRs) represent the largest membrane-bound receptor family in the human genome, consisting of approximately 800 members[1]. GPCRs are expressed throughout the body and play crucial roles in signal transduction by responding to a wide range of external signals, including photons, ions, neurotransmitters, hormones, and proteins[2]. Due to their involvement in numerous physiological processes, GPCRs are linked to various diseases, and nearly 30% of drugs on the market target this receptor family[2]. A majority of such drugs are small molecules or peptides that bind to the orthosteric site that is conserved in sequence and structure. Prominent drug target receptors, including, for example, incretin receptors, dopaminergic, muscarinic, and serotonin receptors, have functionally distinct isoforms with a high degree of sequence similarity in the orthosteric site.

The non-conserved structural regions of highly homologous receptors are often located in disordered or unstructured loop regions that are dynamic. The functional relevance of the disordered regions often emerges from their dynamic ensemble of states. Hence, the realizable therapeutic efficacy of many GPCR therapeutics directed at specific isoforms is limited due to a lack of target specificity. To address this challenge, we investigate isoform-specific information embedded in the structural dynamics of GPCRs.

GPCRs are dynamic proteins, and in response to agonist and trimeric G-protein binding, they undergo structural transition from inactive to active states, leading to activation of the G-protein triggering downstream signaling. While the distinct conformational states of GPCRs, such as inactive, active, and intermediate states, have been

[1]Department of Computational and Quantitative Medicine, Beckman Research Institute of the City of Hope, Duarte, CA, USA. [2]Irell and Manella Graduate School of Biological Sciences, Beckman Research Institute of the City of Hope, Duarte, CA, USA. [3]Department of Genetics, Cell and Developmental Biology, University of Minnesota, Minneapolis, MN, USA. [4]These authors contributed equally: Babgen Manookian, Elizaveta Mukhaleva. ✉e-mail: nvaidehi@coh.org; arodin@coh.org; sbranciamore@coh.org

identified using high-resolution cryo-electron microscopy and X-ray crystallography[3,4], characterizing the transitions among these states at an atomic level using NMR, single-molecule FRET, HDX, and DEER is challenging due to the lack of atomic-level spatial and temporal resolution of these measurements for membrane proteins. Hence, very often the results from these experimental techniques are integrated with molecular dynamics (MD) simulations to study GPCR activation and deactivation processes at an atomic level[5]. Valuable insights have been gained on residues that undergo conformational changes between active and inactive state ensembles in several GPCRs using MD simulations and their analyses[6–10].

Since long-time scale MD simulations are accessible, MD simulations that recapitulate large-scale state transitions have become feasible. The strength of MD simulations lies in their ability to provide detailed atomistic-level information on the ensemble of states and the transitions between them. While MD has emerged as a complementary tool to dissect GPCR dynamics, the sheer volume of data in multi-scale MD simulations requires novel, scalable, and unbiased strategies to reveal structural insights into residue communities that regulate conformational transitions.

The recent wave of artificial intelligence (AI) and machine learning/deep learning (ML/DL) has led to great progress in novel applications to protein structure prediction[11] and dynamics[12,13]. Developments in ML, including neural networks[14] and more sophisticated forms of non-DL predictors[15,16], have provided new toolkits for the task of investigating and predicting protein dynamics. Generative AI-based methods are currently being developed for enhanced sampling of protein conformational space[17,18]. Despite the predictive potential of these ML methods, their lack of interpretability is a limiting factor in applying them for gaining granular mechanistic insights into protein dynamics. Additionally, there is no time-resolved ML model that can be used to analyze the MD simulation trajectories to identify the residues that enable conformational transitions in proteins. In general, there is a lack of interpretable, fully data-driven time-resolved ML methods that unveil patterns in the protein dynamics data at the atomistic scale.

Bayesian network model or BNM (a prominent class of probabilistic graphical models) is an interpretable unsupervised ML model that has proven to be effective in identifying even weak significant probabilistic relations within noisy data. BNM has been successfully applied to analyze diverse biomedical datasets[19–23]. Recently, we have successfully applied BNM in analyzing MD simulation trajectories of GPCR: G-protein complex dynamics[24], and identified cooperative residue contacts in the GPCR:G protein interface that facilitate selective G-protein coupling[24]. However, currently, there is no fully data-driven, interpretable time-resolved ML approach to analyze time-dependent transition events embedded in MD trajectories that recapitulate the macro-state transitions. Building upon this BNM framework, we have developed an innovative method to map the probabilistic relations among GPCR contacts as a function of time. Here, we introduce the Dynamically Resolved Universal Model for BayEsiAn network Tracking (DRUMBEAT), which enables us to identify key residue communities with coordinated movements that enable and effect macro-state transitions, such as GPCR deactivation.

In this study, we first demonstrate the ability of DRUMBEAT to recapitulate previously known experimental and evolutionary insights into the deactivation process of GPCR structural prototype, the β$_2$-adrenergic receptor (β$_2$AR). In a fully data-driven and unsupervised learning mode, we inferred that the community of residues centered around the contact between C327[7.54] and F332[8.50] enables the deactivation transition to the inactive state, and the transition is completed by the ionic lock contact between transmembrane helices 3 and 6 (TM3-TM6) R131[3.50] and E268[6.30]. While these important transition enabler and effector residue contacts are highly conserved among paralogs of β$_2$AR, they are not strictly conserved among adrenergic receptor family subtypes. Our study of deactivation mechanisms of the two highly homologous D$_2$ and D$_3$ dopamine receptors (D$_2$R and D$_3$R) showed that two residue communities in the extracellular and intracellular regions that are involved in the D$_3$R deactivation mechanism are D$_3$R-specific and not involved in the D$_2$R deactivation. This proves that DRUMBEAT is useful for identifying target-specific residue communities that can be harnessed for isoform-specific modulator design.

## Results

The input to DRUMBEAT is properties extracted from MD simulation trajectories that show macro-state transitions. For the β$_2$AR system, we obtained 14 MD trajectories of deactivation starting from the active state structure, each 2μs to 10μs long, from DE Shaw and associates[25]. For the dopamine receptors D$_2$R and D$_3$R, we performed all-atom MD simulations starting from the agonist and G-protein bound active state structure of D$_2$R (PDB: 8IRS)[26] and D$_3$R (PDB ID: 8IRT)[26]. We removed the agonists and the G proteins and performed 31 runs of MD simulations each extending to 1.1–1.4 μs for D$_3$R and 35 runs extending to 300–600 ns for D$_2$R. The simulation times are different for D$_2$R and D$_3$R since the deactivation transitions occurred at different times. The MD simulations were performed in a POPC lipid bilayer at constant temperature (300 K) and volume using GROMACS2022.1 and CHARMMff v36m as detailed in the "Methods" section. To ascertain if the trajectories show transitions from active to inactive states, we calculated the inter-residue distances that are the hallmark of activation in class A GPCRs. The distance between residues R3.50-T6.34 and Y5.58-Y7.53 (using GPCR database residue numbering system) was calculated for each trajectory in every GPCR system and plotted as 2D density plots shown in Supplementary Figs. 1 and 2. Trajectories 1–11 for D$_2$R and 1–12 for D$_3$R did show a transition from active to inactive state as indicated by the decrease in the distance between the residues R3.50 and T6.34 on transmembrane helices 3 and 6 (TM3 and TM6) and a simultaneous increase in the distance between Y5.58 and Y7.53 in TM5-TM7.

### Dynamically Resolved Universal Model for BayEsiAn network Tracking (DRUMBEAT) applied to analyze temporal transition events in proteins

The details of DRUMBEAT methodology can be found in the "Methods" section as well as the GitHub (www.github.com/bandyt-group/drumbeat), which also provides a demo for the code. The DRUMBEAT method developed in this work consists of three steps as described in the schema in Fig. 1. The first step is to extract protein dynamics data such as pairwise residue contact information from MD simulation trajectories as the input for a ML model (Fig. 1A). Using the MD-specific BNM software BanDyT[27] we generate a BNM universal graph that yields the probabilistic dependencies between the input variables (residue contacts in this study). In this universal graph, the residue contacts are the nodes and the edges are direct, non-transitive dependencies between pairs of contacts in the probabilistic space (Fig. 1B). Finally, as shown in Fig. 1C, the universal graph is contiguously re-parametrized or "rescored" along each trajectory using mutual information (MI), and the high-ranking weighted degree measures are used to pinpoint critical events associated with state transitions in the protein. In principle, the method reported herein can be applied to any temporal or dynamic protein data, and the first step may be augmented with different modalities depending on the nature of the available input data.

To build the BNM universal graph, we collectively analyze the aggregated 14 β$_2$AR deactivation trajectories starting from an active state[25]. Each snapshot from MD simulations is expressed as a set of inter-residue contacts, assigning a value of 1 if the contact is formed, and 0 otherwise (Fig. 1A). The choice of contacts as features is motivated by the fact that structural transitions in GPCRs, such as activation and deactivation, are actuated by cooperative changes in internal residue contacts. Analyzing co-dependencies among residue contacts

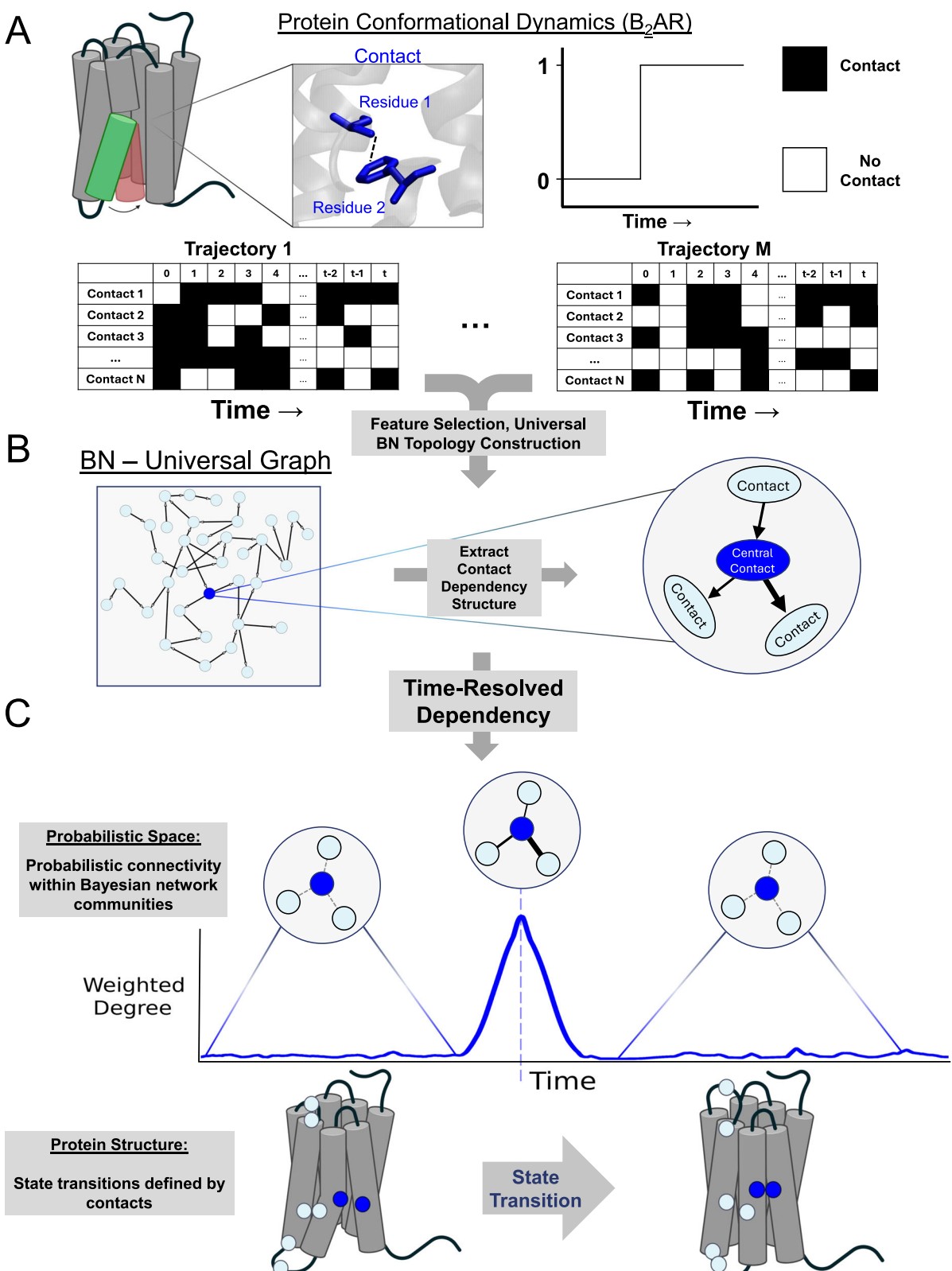

**Fig. 1 | Workflow schema of DRUMBEAT. A** Protein dynamics data serve as an input for the analysis pipeline. MD simulations data were converted to a flat binary residue contact dataset using the GetContacts algorithm. Created in BioRender. Mukhaleva, E. (2025) https://BioRender.com/xayxv90. **B** All trajectories were aggregated, with representative sampling and filter-based feature selection, to create a universal graph via Bayesian network construction. The resulting graph is an interpretable probabilistic model that covers the transition of β2AR from active to inactive state. Embedded feature selection resulted in the identification of key probabilistic communities of contacts for further analysis. **C** A time-resolved Bayesian network was used to analyze transitions between protein conformations along the trajectories by continuously evaluating weighted degrees associated with key communities. Created in BioRender. Mukhaleva, E. (2025) https://BioRender.com/druo1p3 and https://BioRender.com/uo3axod.

will thus enable the understanding of allosteric modulation mechanisms across distant regions[28–32]. Using the matrices of contacts from multiple trajectories, we generate a BNM universal graph that comprehensively shows the significant direct local and non-local dependencies between residue contacts across the ensemble of trajectories.

Nodes in this BNM represent residue contacts, and the edges reflect the direct, non-transitive dependencies between pairs of contacts in the probabilistic space (Fig. 1B). Some nodes show more connectivity over others, approximating a scale-free node connectivity distribution–not uncommon with biological BNMs[33] (Supplementary Fig. 3). We quantified the connectivity of a node by its weighted degree ($D_i$) which is the sum of all edge strengths connecting to a node $i$. Nodes with high weighted degree reflect contacts with the highest impact on the GPCR dynamics due to their probabilistic relations with many other contacts. In our previous work, we showed that nodes with high weighted degree represent cooperative contacts that allosterically regulate G-protein coupling[24].

Although the BNM universal graph is built from temporal data, the model does not explicitly have a temporal component. In other words, the connectivity in the network and corresponding strengths are a superset of all possible direct dependencies across the entire corpus of data (hence "universal graph"). We therefore incorporate a time-resolved weighted degree, $D_i(t)$, which measures the changes in probabilistic dependencies of individual contacts over time (Fig. 1C). To identify residue pairs that enable and effect the transitions we consider pairs of contacts given by the edges in the universal graph and continuously compute their dependency over time via MI in a time window that scans across each trajectory (more details in "Methods"). Each contact in the network is associated with an immediate probabilistic neighborhood, a **T**ime-**R**esolved **A**llosteric **C**ommunity or TRAC, where the strength of connectivity is temporally resolved and is given by $D_i(t)$ (Fig. 1C). We find that $D_i(t)$ of individual TRACs vary over time with regions of the trajectory where $D_i(t)$ is zero or very small, suggesting little or no dependency between nodes in the TRAC. Conversely, there are specific times at which $D_i(t)$ peaks, suggesting a moment of synchronous and cooperative movement of the constituent contacts of the TRAC in question. It is important to note here that the peaks in $D_i(t)$ do not correspond to distinct protein conformational states but rather serve as a signal denoting a transition between two states. As we show in the following sections, these contacts can mediate and/or facilitate the transitions that occur during the deactivation of the receptor. In subsequent sections, we will analyze the behavior of these TRACs over time in multiple trajectories and identify, in a fully data-driven manner, a small subset of TRACs whose contacts and residues are explicitly important in receptor dynamics and conformational transitions.

### Application of DRUMBEAT to the deactivation of β₂-adrenergic receptor

We utilized 14 MD simulation trajectories of β₂AR deactivation (with varying lengths ranging from 2 to 10 μs), all of which contained a transition from an active (nanobody-bound active state structure of β₂AR) state to the inactive state[25]. To derive the BNM universal graph, each trajectory was encoded as a set of binary streams of temporally varying 1500–1700 unique inter-residue contacts as given by the program GetContacts[34] (example contacts represented by black lines in Fig. 2A). To achieve a computationally tractable dataset for BNM convergence, we omitted contacts between neighboring residues and those having low MI with other contacts, retaining around 1000 contacts per trajectory (Fig. 2A, details in "Methods"). We then concatenated the individual binary datasets and further reduced the number of frames using representative subsampling, leading to a final dataset with 14,000 frames and 590 contacts. The resulting BNM universal graph derived from this dataset (590 nodes and 1390 edges representing direct dependencies between contact pairs) comprehensively describes the dependency structure among contacts for all the trajectories (Fig. 2B, Supplementary Data 1). We highlight two of the contacts in the BNM universal graph showing the highest weighted degrees ($D_i$), R131³·⁵⁰_E268⁶·³⁰ and C327⁷·⁵⁴_F332⁸·⁵⁰, colored in blue and purple, respectively. The contact R131³·⁵⁰_E268⁶·³⁰ was previously shown to be a critical structural element or a macroswitch that undergoes significant conformational change upon activation[28,31]. This serves as initial evidence of the method's ability to identify contacts important for the transition.

### Analysis of the ensemble of β₂AR transition trajectories reveals a subset of contacts as mediators of transition

To derive the time dependencies of each contact, we calculate its weighted degree $D_i$, as a function of time for individual trajectories, using the BNM universal graph as a common topology (details in "Methods"). By monitoring the time evolution of the weighted degree of each contact in individual trajectories, as described in the previous section, we explicitly determine when these probabilistic dependencies manifest during a trajectory and infer the protein dynamics during transition.

For each separate trajectory, the computed $D_i(t)$ for each node yields a series of peaks of varying size associated with the manifestation of the probabilistic neighborhood at various times throughout the trajectory. Figure 2C shows the results for three trajectories of β₂AR (see Supplementary Fig. 5 for all 14 deactivation trajectories of β₂AR). For each trajectory, $D_i(t)$ is plotted for the top 50 contacts ranked by amplitude. For reference, indicative of the deactivation process in GPCRs, the interhelical distance between TM3 and TM6 helices is also plotted in gray. We identified a subset of the key contacts by applying three criteria: (1) signals with amplitude above one bit, or about one standard deviation, (2) occurrence of the same signal in all the trajectories, and (3) no more than one or two peaks in a single trajectory. The third criterion arises from our goal of analyzing macro-state transitions in the protein. Here, the two key signals remain (Fig. 2C)–the blue signal corresponds to a set of contacts with concentric peaks, and the purple signal corresponds to a single contact and manifests at the transition defined by the drop in interhelical distance. The remaining signals, shown in brown, have less reproducible features across the trajectories, and although they might contain potentially interesting and useful information on microstate transitions within the active or inactive state ensembles, they will not be the focus of this study.

To select the contacts in a fully data-driven manner, we ranked contacts based on the amplitude of weighted degree $D_i(t)$, as well as prevalence in multiple trajectories for all TRACs. Figure 2D shows the contact ranking in the form of a heat map of the amplitude of weighted degree $D_i(t)$, where the tan squares denote contacts with a weighted degree above the threshold and brown squares otherwise. A large proportion of contacts failed to meet the threshold in even a single trajectory, reiterating the power law distribution for connectivity of contacts in this system. On the right, Fig. 2D shows a zoom-in of the contact ranking for the top 50 contacts and highlights the contacts associated with the enabler and effector signals with the corresponding blue- and purple-colored boxes, respectively. The top-ranking contacts by these criteria are C327⁷·⁵⁴_F332⁸·⁵⁰ and R131³·⁵⁰_E268⁶·³⁰, and we will focus on these TRACs to identify the mechanisms by which the receptor deactivates.

When analyzing the signals in all the trajectories (Fig. 2C and Supplementary Fig. 5), we found that the contacts C327⁷·⁵⁴_F332⁸·⁵⁰ and R131³·⁵⁰_E268⁶·³⁰ (blue and purple signals, respectively) always appear, and, interestingly, the peak of the C327⁷·⁵⁴_F332⁸·⁵⁰ contact always precedes the peak corresponding to R131³·⁵⁰_E268⁶·³⁰. Thus, we infer that the TRAC C327⁷·⁵⁴_F332⁸·⁵⁰, along with other concentric signals, are the enablers, and the TRAC R131³·⁵⁰_E268⁶·³⁰, the purple signal, is the effector of the transition from active to inactive states.

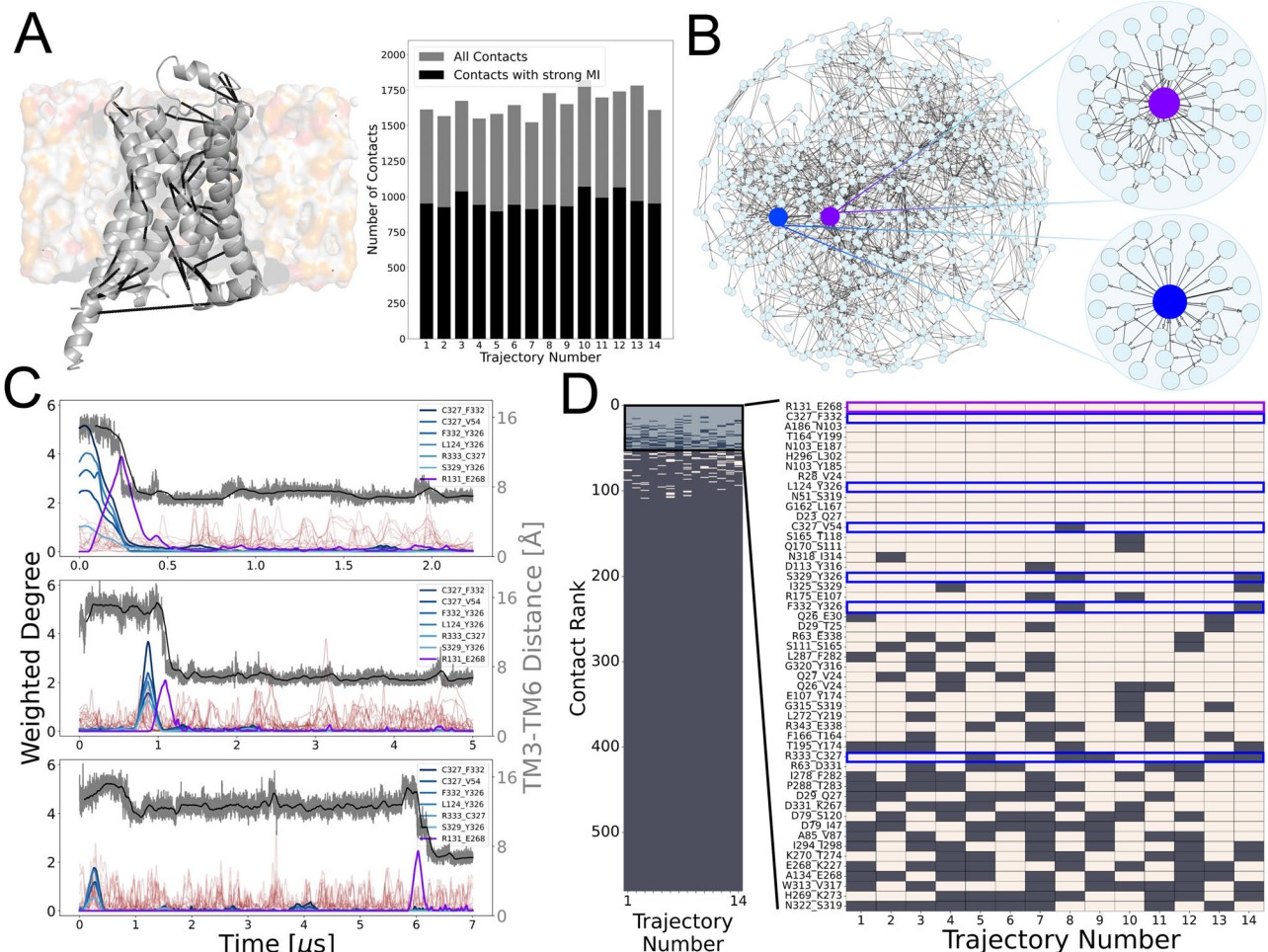

**Fig. 2 | Construction of the universal graph from an ensemble of trajectories, and subsequent time-resolved Bayesian network output. A** Dynamics data in the form of MD trajectories of β₂AR in lipid membrane transitioning from active to inactive state. Universal dataset built using contacts with strong mutual information with other contacts. **B** Universal graph output from the Bayesian network modeling, and corresponding time-resolved allosteric networks shown with the central node R131_E268 and C327_F332 highlighted in purple and blue, respectively.

**C** Time-resolved Bayesian network output for trajectories 8, 11, and 13 with main contacts highlighted in purple and blue. The remaining contacts are shown in brown. The interhelical distance between TM3 and TM6 is plotted in gray with the corresponding simple moving average in black. **D** Ranking of contacts by weighted degree above the threshold of 1 bit. The contacts shown in (**C**) are highlighted, also in purple and blue. Source data are provided as a Source data file.

## The transition between conformational states in β₂AR as seen via a change in residue community connected to enabler C327⁷·⁵⁴_Y326⁷·⁵³ and effector R131³·⁵⁰_E268⁶·³⁰ contacts

Figure 3A shows the probabilistic neighborhoods associated with the enabler C327⁷·⁵⁴_Y326⁷·⁵³ (blue) and effector R131³·⁵⁰_E268⁶·³⁰ (purple) TRACs. The concentric nature of the peaks that make up the enabler signal (Fig. 3A; blue) is represented by a strongly interconnected probabilistic neighborhood with multiple overlapping residues. Specifically, the residues C327⁷·⁵⁴ and Y326⁷·⁵³ occur in multiple enabler contacts and have been shown by experiments to be important for the receptor function, with mutation in these residues resulting in increased or decreased agonist-mediated β₂AR activity[35–38]. Moreover, Y326⁷·⁵³ is a part of the well-characterized and conserved NPxxY motif that acts as a structural toggle, playing essential roles in rearranging the intracellular region for downstream signaling[28,29,32,39,40]. Second, and somewhat less obvious, is the closeness in probabilistic space between the two neighborhoods. There exists a set of 10 connector contacts that have an enabler and effector central node as an immediate neighbor. These connector contacts play a distinct and informative role in the conformational changes attributed to the enabler and effector signals, as highlighted in this section. In addition, this degree of closeness in the

probabilistic space between the enabler and effector TRACs (although distant in structural space) implies a strong probabilistic allosteric relationship between the two conformational changes that they represent. Interestingly, the closeness in probabilistic space does not definitively imply temporal or spatial closeness of the signals, as we highlight in an example below.

As mentioned above, each TRAC signal represents the coordinated motion of a subsection of the protein structure defined by the constituent contacts in the respective TRAC. We declare these events of coordinated motion as transitions between distinct conformational states of the protein. Consider the trajectory shown in Fig. 3B where there is an enabler signal and an effector signal. The presence of these two signals implies that two transitions are occurring in the trajectory between three distinct states. To further explore this idea, we carried out a principal component analysis (PCA) using the 98 contacts in the combined enabler and effector network in Fig. 3A. Figure 3B, C depicts the PCA results for two different trajectories and both show the presence of three states in the state space defined by principal component 1 (PC1) and principal component 2 (PC2). Moreover, the contacts with the highest weighting in PC1 overlap with the set of enabler contacts, while the highest weighted contact for PC2 is the effector central contact. This suggests that the enabler signal represents the transition

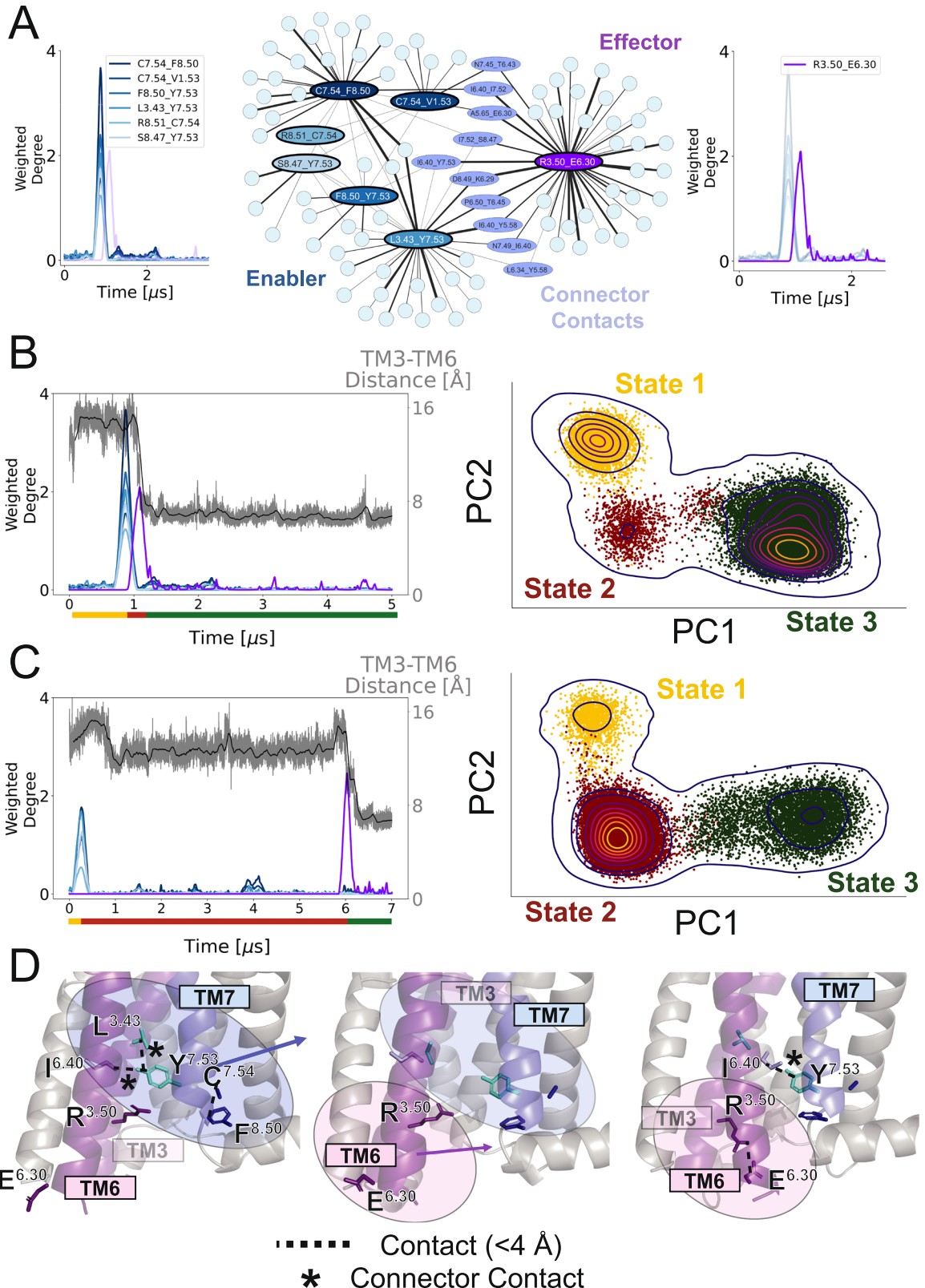

**Fig. 3 | Analysis of "enabler" and "effector" signals is shown in blue and purple, respectively. A** Subnetwork corresponding to the enabler and effector TRACs with corresponding central nodes. The subnetwork shows a set of contacts that connect the two TRACs labeled as the "connector" nodes. **B**, **C** Conformational space analysis using two different trajectories and the contacts in the subnetwork. Principal component analysis reveals three distinct states in both trajectories. Coloring of the points in the PCA space with respect to the time points given by the peaks in the trajectory shows the blue peaks corresponding to the transition between State 1 and State 2, and the purple signal—between State 2 and State 3. The interhelical distance between TM3 and TM6 is plotted in gray with the corresponding simple moving average in black. **D** Structural analysis of the three states taken from the trajectory in part C. Transition from State 1 to State 2 is characterized by the movement of TM7 away from TM3, resulting in the breaking of contacts in the enabler TRAC. The subsequent transition from State 2 to State 3 is characterized by TM6 moving in towards TM3, resulting in the formation of the salt bridge between effector TRAC central contact residues R131$^{3.50}$ and E268$^{6.30}$. Source data are provided as a Source data file.

between state 1 and state 2, and the effector signal represents the transition from state 2 to state 3.

We can also analyze the temporal relationship between the two signals, and therefore the transitions. In principle, the population of individual states in the PCA space correlates with the time spent in that state as recapitulated by the MD simulations. The peaks provide a quantitative measure of the temporal adjacency of the state transitions, and therefore the states themselves. For instance, in Fig. 3B, the enabler and effector signals are within 1 µs of each other. This ultimately corresponds to the low density of state 2 in the PCA analysis on the right. This temporal adjacency coincides with the closeness of the two communities in the network shown in Fig. 3A. However, the trajectory in Fig. 3C shows the enabler and effector signals nearly 6 µs apart, far away in time, and yet, its dynamics still corresponds to the closely connected probabilistic neighborhoods. This implies that even though the conformational changes in the protein occur at different time points, the enabler and effector residue communities remain the same. This is a key finding of this work.

Figure 3D shows the residue contacts involved in the enabler and effector signals in the three distinct states delineated from PCA. In state 1, the central contacts in the enabler TRAC break and make contacts involving residues C327$^{7.54}$ and Y326$^{7.53}$ as TM7 moves away from TM3. During this change, a large portion of breaking and forming contacts between C327$^{7.54}$ and F332$^{8.50}$ also occurs between TM7 and helix 8. This transition gives rise to state 2, an intermediate state between active and inactive conformations, a finding that follows the report[25] from which the trajectories were obtained. The precise structural motion and key residues involved provide a deeper understanding of the transition to the intermediate state. After some time in this intermediate state, the second transition associated with the effector signal occurs, leading to state 3. This transition occurs via TM6 swinging into the G-protein binding pocket towards TM3, ultimately forming a salt bridge between the central contact of the effector TRAC, R131$^{3.50}$, and E268$^{6.30}$, indicating the transition of the receptor into the inactive state[41]. Studies indicate the reverse sequence corresponds to the activation pathway for β$_2$AR[42]. However, this may be unique to β$_2$AR and should not be generalized to all receptors. The DRUMBEAT methodology merely identifies enabler residues of deactivation and does not directly imply equivalence with those involved in activation pathways. We note here that the residue R3.50 could also form a salt bridge with its neighbor D3.49, as seen in the crystal structures of the inactive state of some class A GPCRs[29]. We examined this contact in the β$_2$AR trajectories and found that it does not consistently form upon deactivation, in line with previous reports of significant basal activity for β$_2$AR leading to the lack of formation of the R3.50_D3.49 salt bridge in the inactive state[43]. Notably, the connector contact between residues I278$^{6.40}$ and Y326$^{7.53}$, initially broken during the first transition, is reformed in this state, reiterating its structural role in both transitions. The importance of position 6.40 is further emphasized by the fact that it has been reported to stabilize the inactive state of rhodopsin[44]. Previous studies corroborate the importance of these key residues. For instance, the salt bridge between R131$^{3.50}$ and E268$^{6.30}$ is one of the established indicators of rhodopsin deactivation[45], and R131$^{3.50}$ is part of the conserved ED/RY motif on TM3 across all class A GPCRs[46]. It is worth noting that position 6.30 has variation across different class A GPCRs, leading to the R3.50_E6.30 salt bridge primarily existing in amine receptors and opsins[47]. Also, Y326$^{7.53}$ contributes to GPCR activation via the conserved NPxxY motif[48].

The mechanistic conclusion for β$_2$AR came from a fully data-driven unsupervised learning approach without any previous information or knowledge of the system (except for the interhelical distance, which serendipitously served as a ground-truth positive control). Taken together, our DRUMBEAT-based, fully data-driven analysis has uncovered the two transitions during deactivation and the

precise residues involved in enabling these transitions. We envision these types of findings manifesting from our methodology for any other de novo protein system where no prior information is available.

## Transition enablers and effector residues are conserved across adrenergic receptors

To investigate whether the residues identified as crucial during the deactivation transitions of β$_2$AR are preserved in the adrenergic family, we have analyzed their conservation level in comparison to other residues by computing the conservation score ($C_i$) using Von Neumann entropy[49]. This method considers the chemical similarity between different amino acids to obtain a more reliable evaluation of the degree of conservation along the alignment. Briefly, it constructs the density matrix ρ from the relative frequencies of residues at each position and a similarity matrix (BLOSUM50). The Von Neumann entropy is $S_i = -\sum_i \lambda_i \log_{20} \lambda_i$ with $\lambda_i$ the eigenvalues of ρ. Then, the conservation score is obtained by subtracting $S_i$ from the maximum possible entropy (in our case, 1), $C_i = 1 - S_i$. Using a multiple sequence alignment (MSA) of the 68 adrenergic receptor family sequences from 20 species from the GPCR database (https://gpcrdb.org/), we computed $C_i$ for each position of β$_2$AR in the alignment (Fig. 4A). As evident from Fig. 4A, B, key residues involved in the transition have higher conservation level compared to other receptor residues indicating their implication in β$_2$AR stability and function. Numerous prior studies have demonstrated the importance of the residues identified by DRUMBEAT as enablers and effectors of transition. The residue R131$^{3.50}$ is part of the conserved E/DRY motif and is responsible for modulating agonist binding and G-protein coupling and governing receptor conformations[46,50]. In the CWxP motif, the residue P288$^{6.50}$ is involved in the rotamer toggle switch mechanism that plays a role in active forms of GPCRs[51,52]. Moreover, the connector residues N322$^{7.49}$ and Y326$^{7.53}$ are part of the conserved NPxxY motif, which contributes to GPCR activation and state transition[39,48]. The probability distribution function of the conservation score in Fig. 4 shows that the amino acid positions that are part of the enabler and effector central contacts(red), connector contacts that enable direct "communication" between those hubs in the probabilistic space(green), and the set of all the contacts in the first neighborhood of the main hub(yellow) are highly conserved compared to all the residues in the receptor. The rank-sum test shows a statistically higher level of conservation (higher values of $C_i$) in DRUMBEAT-identified positions, indicating that evolutionary selection pressures are congruent with the relationships identified by DRUMBEAT.

## Do the transition enabler and effector residues differ among dopamine receptor D$_2$R and D$_3$R subtypes?

Designing subtype-selective modulators and/or ligands targeting highly homologous proteins is one of the foremost challenges in GPCR drug discovery. We applied DRUMBEAT to identify the residue communities involved in the deactivation transitions of D$_2$R versus D$_3$R, two prominent drug targets that so far have not been tractable to isoform-selective modulation[53,54]. Such an example would also demonstrate the broad applicability of the DRUMBEAT framework. We performed deactivation MD simulations on D$_2$R and D$_3$R, generating 11 and 12 trajectory ensembles, respectively, with details provided in the "Methods" section. Preprocessing of the trajectory data included contact mapping and feature selection similar to those described for β$_2$-adrenergic receptor, followed by constructing the BNM universal graph (Supplementary Data 1). We applied DRUMBEAT and identified relevant TRACs for each system.

We find that β$_2$AR, D$_2$R, and D$_3$R exhibit both shared and distinct deactivation mechanisms. Firstly, we found that in the MD simulations, the deactivation of D$_2$R is faster in time scale by an order of magnitude when compared to β$_2$AR and D$_3$R (0.1 vs 1−5 µs). This could be due to D$_2$R being involved in motor control and rapid response modulation,

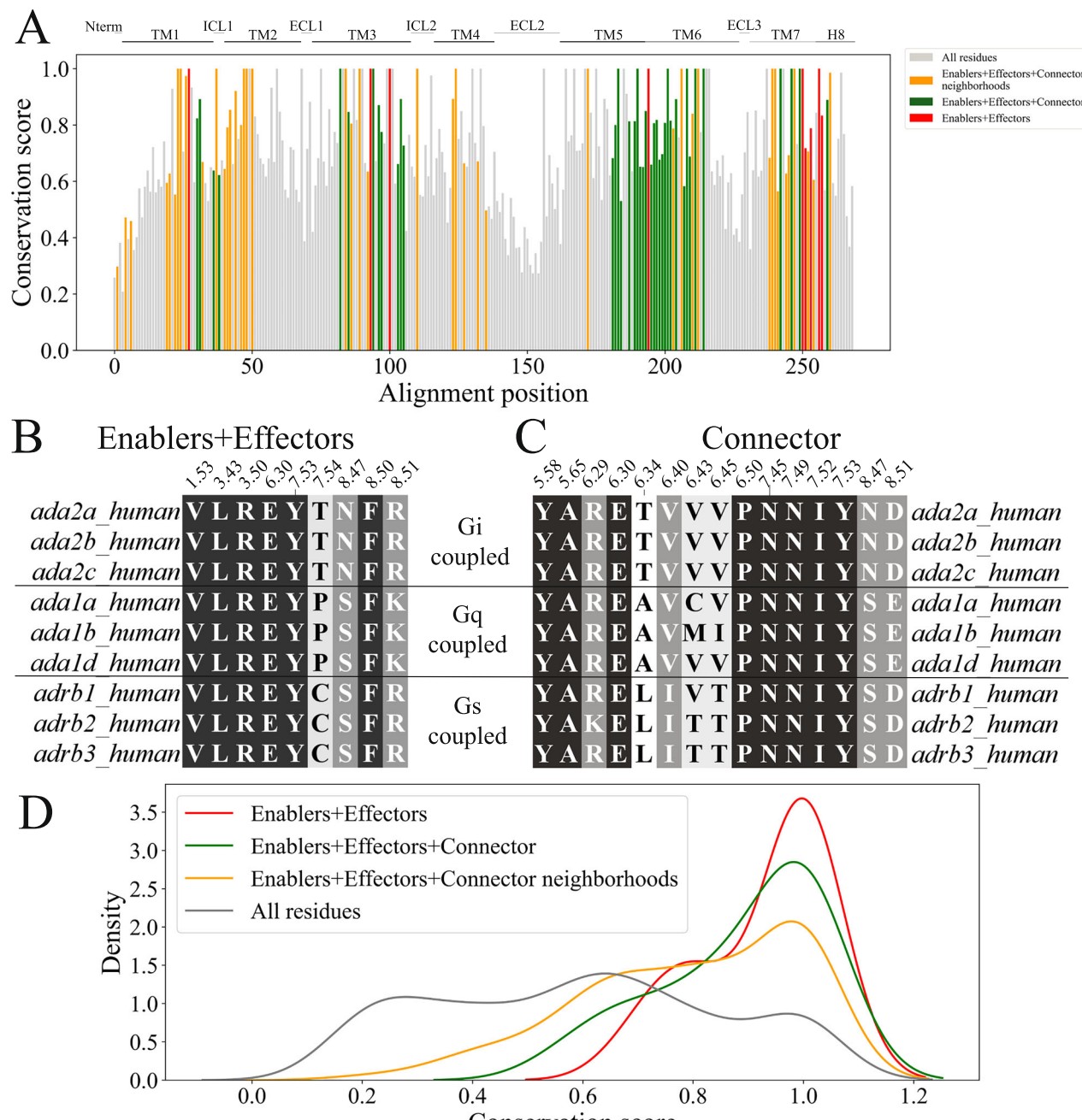

**Fig. 4 | Evolutionary analysis of residue communities involved in conformational transitions in β₂AR. A** Mapping of the conservation score for all residues in human adrenergic receptors alignment. Sets of residues identified via DRUMBEAT are highlighted (yellow, green, and red) and show a higher overall conservation score compared to all residues (gray). **B** Alignment of residues contained in the enabler and effector central contacts across human adrenergic receptors. **C** Alignment of an expanded set of residues that contains the connector contacts between enabler and effector subnetworks. **D** Probability density function of conservation score computed using Von Neumann entropy for enabler+effector central contact residues (red), connector residues (green), entire neighborhood residues (yellow), and all protein residues (gray). Source data are provided as a Source data file.

requiring fast on/off signaling[55,56]. In contrast, D₃R is mainly involved in emotional and motivational states where prolonged signaling may be beneficial[57]. In addition to the biological/physiological explanation, we found that D₂R undergoes deactivation via two key events happening in the receptor, whereas D₃R requires three events. This provides a probabilistic/combinatorial explanation for why D₃R takes longer to deactivate. Although the reported dopamine receptor MD simulations are shorter than β₂AR, we extended the simulation time to ensure the receptor did not go back to intermediate or active states.

The top-scoring residue contacts involved in D₂R deactivation were identified using DRUMBEAT, with the subnetwork shown in

Fig. 5A. Five key contacts (TRACs) arise that span the intracellular and transmembrane regions of the receptor. The $D131^{3.49}$_$T68^{2.38}$ contact at the intracellular region is determined to be a key contact in both D₂R and D₃R. In the transmembrane region, $D80^{2.50}$, $S121^{3.39}$, and $N422^{7.49}$ residues form two key contacts in D₂R that show a hydrogen bond. Most notably M6.36_Y7.53 contact appears as a connector in D2R, which is analogous to the connector contact M6.40_Y7.53 in β₂AR. This overlap highlights the conserved nature of connector contacts across different GPCRs. $D131^{3.49}$_$T68^{2.38}$ contact plays an important role in D₃R deactivation as well, along with 4 other contacts (subnetwork shown in Fig. 5B). Interestingly, the contacts highlighted for the D₃R system span

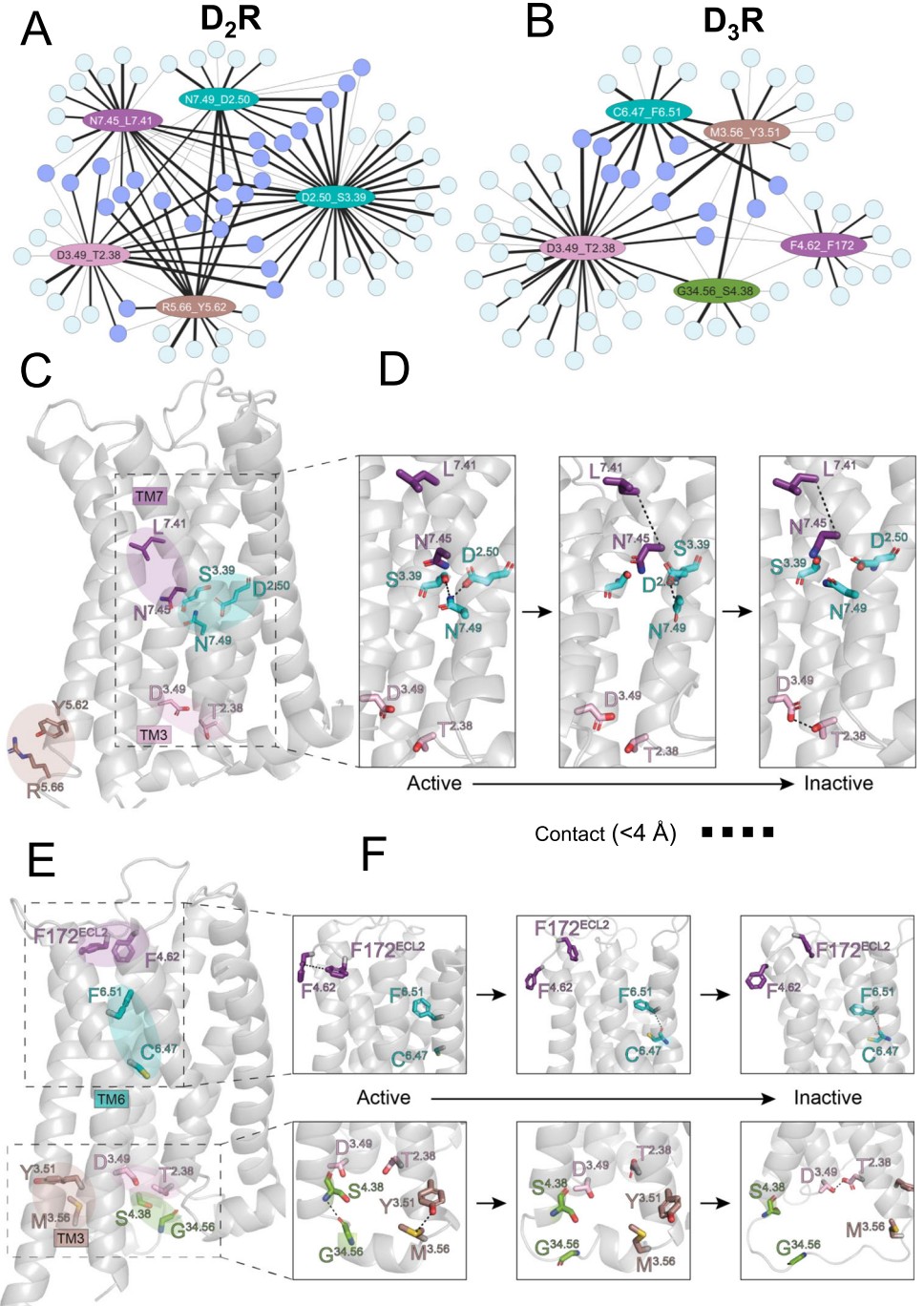

**Fig. 5 | Application of DRUMBEAT to Dopamine $D_2$ and $D_3$ receptors.** Subnetwork from the universal graph for the top five ranked contacts for $D_2R$ (**A**) and $D_3R$ (**B**). Connector contacts shared between top-ranked contacts are highlighted in purple. **C** Projection of top contacts on $D_2R$ structure. **D** Mapping of the structural changes that occur in $D_2R$ via changes in key contacts as identified by DRUMBEAT. **E** Structure of $D_3R$ with key contacts highlighted. **F** Mechanism of deactivation for $D_3R$ via changes in key contacts. Notably, contacts in both intracellular and extracellular regions initiate the transition for $D_3R$. Both receptors transition into the inactive state via the formation of the $D^{3.49}\_T^{2.38}$ contact (shown in orange).

the entire protein (see Fig. 5E). Specifically, the contact $F170^{4.62}\_F172^{ECL2}$ at the top of TM4 and in the ECL2 stands out in the $D_3R$ system. This contact is unique to the $D_3R$ and has been shown via mutation studies to play a significant role in ligand binding, although it is not located directly in the ligand binding site[58]. We hypothesize that it stabilizes the global architecture of the extracellular region via extracellular loop 2, which can influence the shape and accessibility of the binding pocket. The protein-spanning nature of TRACs in $D_3R$ alludes to the allosteric nature of the receptor's activity. This is further corroborated

by the fact that the neighboring contacts in the $F170^{4.62}\_F172^{ECL2}$ TRAC are in both the extracellular and intracellular regions of the receptor (see below in "Discussion"). The difference in sites of the $D_2R$ and $D_3R$ receptor TRACs manifests in distinct mechanisms by which these receptors deactivate.

Even though $D_2R$ and $D_3R$ are highly similar in sequence and structure, DRUMBEAT uncovered distinct mechanisms by which each receptor transitions between active and inactive states. In $D_2R$, initially, the contacts between $D80^{2.50}$, $S121^{3.39}$, and $N422^{7.49}$ residues stabilize

the active state via hydrogen bonding. Notably, this hydrogen bond network is also present in $\beta_2AR$, consistent with the high conservation of these residues across class A GPCRs. In $\beta_2AR$, the network appears to stabilize the active state and is disrupted as the receptor transitions to the inactive state, paralleling observations in $D_2R$. Importantly, the sequence in which the hydrogen bonds break is largely preserved in both receptors: the D2.50–S3.39 hydrogen bond is lost first, followed by disruption of the N7.49–D2.50 bond (see Supplementary Fig. 5). Moreover, these residues (S3.39, D2.50, and N7.49) are known to be a part of the conserved sodium-binding site in Class A GPCRs, which plays a key role in modulating receptor activation and stabilization of the inactive state by binding the sodium ion[59]. The disruption of this hydrogen-bonded network prior to deactivation provides mechanistic insight that links this well-established structural motif to dynamic switching between active and inactive states. The first step in deactivation is mediated by the formation of the N418$^{7.45}$_L414$^{7.41}$ contact at the kink region of TM7 (shown in red in Fig. 5C, D). This TRAC may correspond to the outward movement of TM7 observed in the $\beta_2AR$ study, representing a potential preliminary step in deactivation, though this relationship was not uniformly observed across $D_2R$ trajectories. Then, hydrogen bonding between D131$^{3.49}$_T68$^{2.38}$ forms in the inactive state, possibly stabilizing the conformation. Interestingly, this contact also plays a similar role in $D_3R$ upon reaching the inactive state (see Fig. 5F). The difference in the $D_3R$ deactivation mechanism arises from the other TRACs identified as central contacts. Notably, initial transition out of the active state involves breaking of contacts in both the intracellular and extracellular region of the receptor, suggesting allosteric coordination (Fig. 5F). The contact C341$^{6.47}$_F345$^{6.51}$ at the kink region of TM6 facilitates the final transition into the inactive state, corresponding to the helical movement of TM6 towards TM3, also in line with what we saw for $\beta_2AR$ as well as what is commonly reported for GPCR deactivation. Although the stability of the inactive state in both $D_2R$ and $D_3R$ is facilitated by the D131$^{3.49}$_T68$^{2.38}$ contact, the active state is held together by a notably different set of residues and contacts. These findings point to evolutionary adaptations that likely enable functional specificity, despite overall receptor structure homology.

One final yet notable difference between $\beta_2AR$ and dopamine receptors is the behavior of the salt bridge formed between arginine on TM3 (R$^{3.50}$) and glutamate on TM6 (E$^{6.30}$). In $\beta_2AR$ deactivation trajectories, the salt bridge is consistently open during the active state and forms upon deactivation to the inactive state. Once in the inactive state, the salt bridge remains predominantly formed, with occasional fluctuations between open and closed states. These dynamics led to the contact being identified as a key TRAC by DRUMBEAT (R131$^{3.50}$_E268$^{6.30}$), as discussed above. In contrast, the dopamine receptor deactivation trajectories in this study reveal that the salt bridge forms in only a subset of trajectories (4 out of 11 for $D_2R$; 5 out of 12 for $D_3R$) and at a significantly lower contact frequency. This could imply that the salt bridge is more open in the inactive state(s) of dopamine receptors. To further investigate, we examined crystal structures of $D_2R$ and $D_3R$ in the inactive state ($D_2R$: 6LUQ, 6CM4, 7DFP; $D_3R$: 3PBL) for this salt bridge. We also analyzed the MD simulation trajectories of the antagonist-bound inactive states of $D_2R$ and $D_3R$. The crystal structures show the salt bridge is formed in two of three $D_2R$ inactive states (Supplementary Fig. 7A) as well as in the $D_3R$ inactive state (Supplementary Fig. 7B). In the inactive state MD trajectories, the salt bridge forms in approximately 12% of frames for $D_2R$ and 68% of frames for $D_3R$ (Supplementary Fig. 7C). In the case of $D_2R$, inactive state MD trajectories (initiated from the crystal structure−$D_2R$: 6LUQ) show the salt bridge is formed initially, but then opens and remains so for the remainder of the trajectory. These findings indicate that while the salt bridge can form in the inactive state due to the proximity of TM3 and TM6, it is not essential for maintaining a stable inactive

conformation. This suggests the existence of two distinct inactive states and a dynamic salt bridge between TM3 and TM6[60,61].

## $D_3R$ transition is driven by subtype-specific non-conserved residues

The findings from $D_2R$ and $D_3R$ suggest that while the dopamine receptors share some commonality in deactivation mechanisms, their variability could imply subtype-specific deactivation pathways that can be tapped into for subtype selectivity. To investigate this via evolutionary considerations, we divided residue positions into two sets: the set of "fully conserved" (C) positions and the set of "perfectly specific" (S) positions for a particular dopamine receptor class (e.g., $D_2R$, $D_3R$). As shown in the schematic in Fig. 6A, the "fully conserved" positions remain invariant across all classes, whereas the "perfectly specific" positions uniquely distinguish one class of dopamine receptors from all the others. Purifying and adaptive selection are two prominent mechanisms shaping proteins' evolution. Purifying selection removes harmful mutations from the gene pool and preserves specific residues in crucial positions within the protein, thereby maintaining key functionalities in protein structures. Adaptive selection, on the other hand, favors the fixation of advantageous mutations in the population. This process can lead to the neo-functionalization of paralogous genes, where new functions evolve. Purifying selection is likely the dominant evolutionary force acting on the "fully conserved" positions, as any alteration would be strongly selected against due to its impact on the core functionality of the receptor. In contrast, adaptive selection likely served as the primary evolutionary force acting on the "perfectly specific" positions, favoring those that provide a unique function to a given dopamine receptor class. Therefore, the ratio $\rho = \log_2 |S|/|C|$ provides an insight into the dominant evolutionary force acting on the residue community as identified by the DRUMBEAT analysis (i.e., key residues in the process of deactivation). Specifically, negative values denote receptors with mostly fully conserved positions, and positive values denote receptors with mostly perfectly specific positions.

To evaluate the conservation level of TRAC residues in dopamine receptors, we obtained MSA for all dopamine receptors ($D_1$, $D_2$, $D_3$, $D_4$ & $D_5$) in all species from the GPCR database (www.gpcrdb.org). TRACs with a maximum weighted degree above 2.0 bits were extracted from each trajectory in the ensemble, and the alignment was done using the residues in the union set of contacts. Alignment of residues from $D_2R$ TRACs is shown in Fig. 6C, with $D_2R$ sequences highlighted in green. Similarly, Fig. 6D shows the alignment for $D_3R$ TRAC residues, with $D_3R$ sequences likewise highlighted in green. For positions shown in Fig. 6C, D, we calculated $\rho$ and compared between receptors.

We computed $\rho$ for all receptors in this study to provide a comparison of specificity (Fig. 6). We found that $\beta_2AR$ ($\rho = -1.0$) and $D_2R$ ($\rho = -2.5$) both exhibit negative $\rho$ values, indicating that the key residues for deactivation are strongly conserved across the receptor class. In contrast, the key residues for $D_3R$ ($\rho = 0.5$) activation show a high degree of specificity, with nearly an order of magnitude difference, suggesting that these residues are evolutionarily diverging. Sequence conservation scores for $D_2R$ and $D_3R$ (Fig. 6E, F) further reveal that while central contacts in $D_2R$ are highly enriched in conservation, those in $D_3R$ display only a modest increase, underscoring the subtype-specific nature of the $D_3R$ residues. To summarize, our analysis reveals that the key residues involved in $D_3R$ activation are characterized by positions that are evolutionarily diverging within the dopamine receptor family. Critically, this suggests that during evolution, $D_3R$ began to diverge from its otherwise similar $D_2R$ counterpart by developing a unique activation pathway. Our DRUMBEAT-enabled ability to functionally characterize residues at these divergent positions provides direct insight into the functional mechanisms and evolutionary forces driving sub-functionalization, potentially inspiring novel drug molecule designs.

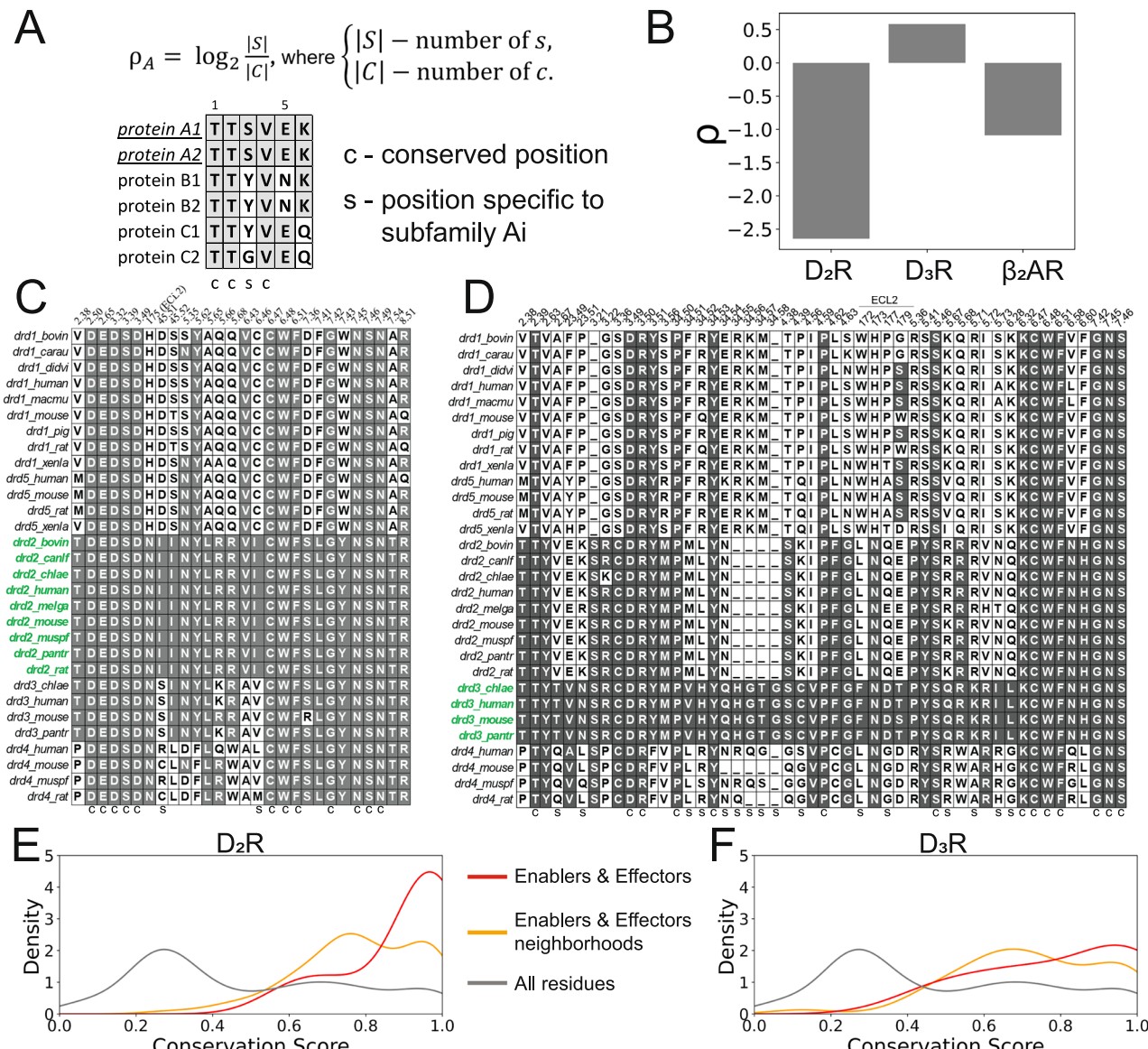

**Fig. 6 | Evolutionary analysis of Dopamine receptors in accordance with findings from DRUMBEAT analysis. A** Quantification of conservation and specificity for residues. **B** Computed value for the ratio ρ for $D_2R$, $D_3R$, and $\beta_2AR$. **C, D** Sequence alignment for $D_2R$ and $D_3R$. **E, F** Conservation score for central contacts (red), central contacts and corresponding neighborhoods (orange), and all residues (gray). Source data are provided as a Source data file.

## Swapping time-resolved allosteric contacts confirms the predicted determinants of receptor deactivation dynamics

To evaluate whether the TRACs we identified using TRBNM for $D_2R$ and $D_3R$ represent critical determinants of the deactivation transition and whether altering them can modify the transition patterns, we performed reciprocal TRAC-swapping mutations between the two receptors at positions where the TRACs differ (Fig. 7A). Specifically, we mutated residues F172/L174 in ECL2 (TRAC F4.62_F172 in $D_3R$), position 5.66 (TRAC R5.66_Y5.62 in $D_2R$) by introducing an R→K mutation in $D_2R$ and a K→R mutation in $D_3R$, and position 34.56 in ICL2 (TRAC G34.56_S4.38 in $D_3R$) by inserting a residue in $D_2R$ and deleting it in $D_3R$ (Fig. 7B, Supplementary Table 1). We conducted 25 deactivation MD simulations for each chimera, obtaining 19 successful deactivation trajectories for the $D_{2/3}R$ mutant and 14 for the $D_{3/2}R$ mutant. In both cases, the time required for deactivation differed significantly from their respective wild-type receptors (Fig. 7). The average deactivation time for the $D_{2/3}R$ chimera increased from 70 to 220 ns, whereas for the $D_{3/2}R$ chimera, it decreased from 330 to 180 ns.

To quantitatively assess these differences, we performed Mann–Whitney $U$ tests (Supplementary Table 2). The results confirmed significant differences between wild-type receptors and their respective mutants, with $U$ statistics of 2 ($p < 0.0001$) for $D_2R$ vs $D_{2/3}R$ and 41 ($p = 0.0283$) for $D_{3/2}R$ vs $D_3R$. Interestingly, not only did the mutations alter the deactivation time scales, but the mutants also behaved in a manner opposite to their respective wild-type receptors. Specifically, the $D_{2/3}R$ chimera exhibited a deactivation time distribution more similar to $D_3R$ than to $D_2R$ ($U = 65.5$ vs $D_3R$ and $U = 2$ vs $D_2R$), whereas the $D_{3/2}R$ chimera behaved closer to $D_2R$ than $D_3R$ ($U = 30.5$ vs $D_2R$ and $U = 41$ vs $D_3R$). These findings indicate that the correct combination of TRACs is required to ensure efficient progression through the deactivation pathway and that swapping TRACs reverses the receptor's deactivation characteristics.

## Discussion

One of the major challenges in pharmacology is identifying target-specific drugs, especially for highly homologous proteins. To achieve target specificity, GPCR drug discovery paradigms in the last decade

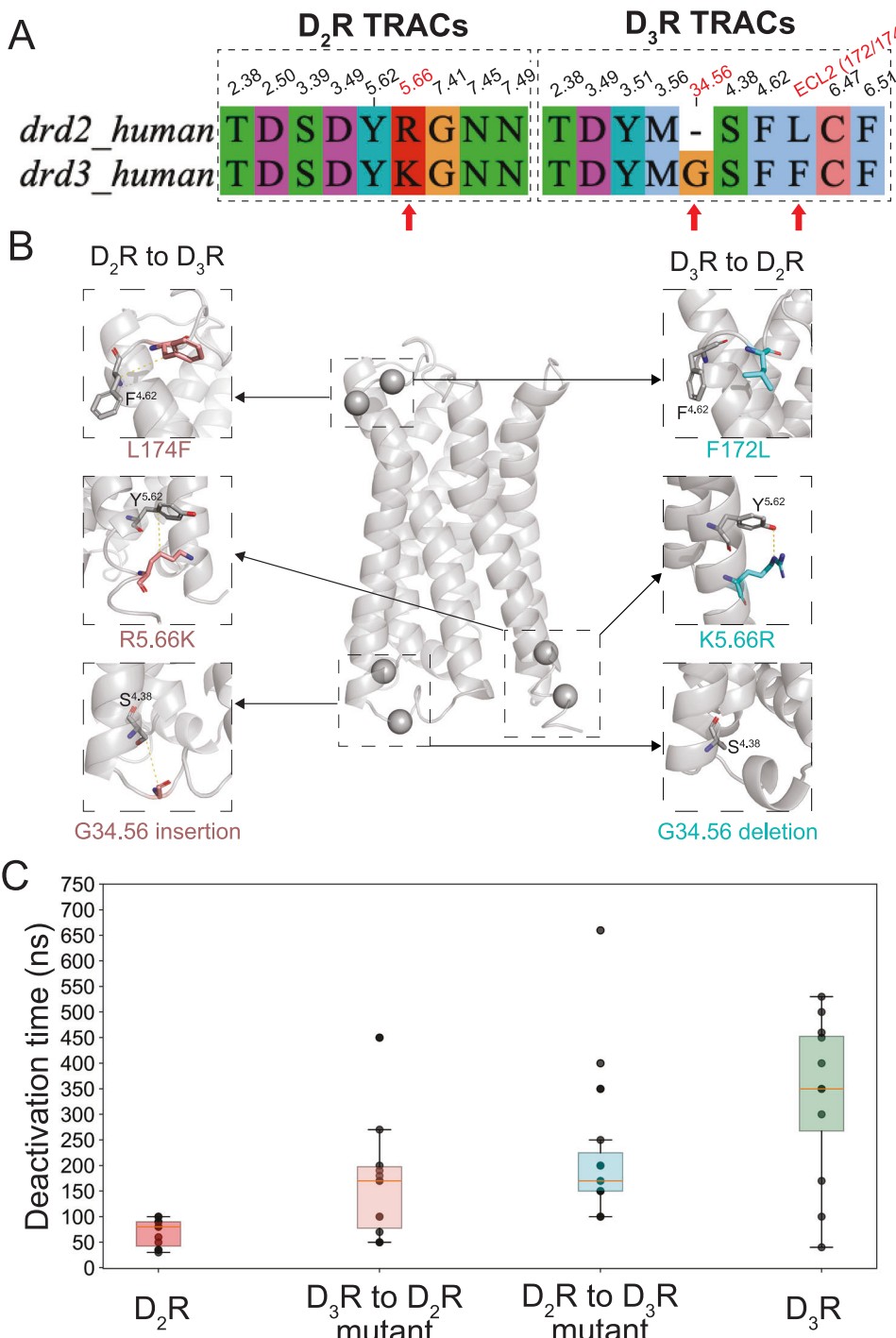

**Fig. 7 | DRUMBEAT-guided mutations in D₂R and D₃R alter deactivation times.**
**A** Alignment of residues in D₂R and D₃R TRACS as identified by DRUMBEAT. Red arrows show the positions where mutagenesis was done. **B** Locations of mutations in the receptor. **C** Box plot of deactivation times for wild-type and mutant D₂/D₃ receptors (nm). D₂R−11 replicates, D₃R−12 replicates, D₂/₃R−19 replicates, D₃/₂R−14 replicates. A two-sided Mann−Whitney *U* test was performed to establish a significant difference between sets (statistics are provided in Supplementary Table 1). Source data are provided as a Source data file.

have pivoted to allosteric modulators[62]. However, understanding the functional impact of allosteric modulators on GPCR signaling is inherently limited by the dynamic nature of GPCR activation. Further, for multiple high-value drug target receptor families, including incretin, dopamine, serotonin, and muscarinic, isoform-specific residues are dispersed across the receptor structure, complicating traditional structure-based strategies to realize isoform selectivity of small molecule therapeutics. In this study, we use an interpretable ML technique (DRUMBEAT) to identify, in a fully unsupervised learning mode, spatially

disperse, non-conserved residue communities that coordinate key conformational transitions in GPCRs. The distinct transition dynamics manifested by D₂R and D₃R provide a structural framework to identify selective allosteric sites and modulators for mitigating the opioid crises (D₃R) and neuropsychiatric disorders (D₂R)[54].

This work addresses a critical challenge in protein dynamics: identifying key regions and residues, or "allosteric residue communities," that facilitate transitions between conformational states and ascertaining how these communities behave temporally. The

DRUMBEAT methodology developed in this study uncovers the temporal dynamics and mechanistic underpinnings of conformational transitions in the $\beta_2AR$ and two highly homologous and sought-after drug targets, namely dopamine $D_2R$ and $D_3R$ receptors. We identified TRACs that correspond to key sub-regions of the protein affecting the conformational changes and how their manifestations coincide with the transition of the receptor from active to inactive or vice versa. In contrast to "microswitches," which are defined as residues/residue pairs that show local conformational changes in the static structures of the active state in comparison to the inactive state, this study identifies residue pairs that enable conformational changes but do not necessarily show conformational changes. Although DRUMBEAT may identify microswitches, it can also uncover new residues or regions of the protein that are critical for activation, but do not fall in the category of microswitches. The results emphasize how probabilistic connectivities can identify critical contacts and their allosteric coordination during key transition events, providing a fresh perspective on receptor transition.

Two dominant computational approaches for studying protein dynamics are dynamic network analysis (DNA)[63] and DL models[11,64,65]. DNA provides interpretable insights by constructing networks where nodes represent residues and edges are indicative of physical proximity. However, this method is inherently local, limiting its capacity to capture long-range allosteric interactions, especially those spanning spatially distant but functionally correlated regions. Conversely, DL methods offer high predictive performance[11] and scalability[64] but often lack interpretability as they provide, at most, feature importance rankings without mechanistic or interaction insights. Additionally, DL tends to rely heavily on extensive prior data, likely requiring hundreds if not thousands of MD trajectories in our case, and often implying transfer learning with possible irrelevant false positives and hallucinations. The DRUMBEAT approach introduced in this study bridges these gaps, combining the interpretability of DNA with the scalability and expressiveness of DL while offering a distinct capability: time-resolved mechanistic analysis. A standout feature of this work is the temporal resolution afforded by DRUMBEAT, which reveals not just the key contacts and their involvement in transition but also the granular temporal anchoring and order of events during transitions. The temporal resolution allows for identifying enabler and effector residue communities, key sub-regions of the protein involved in the deactivation process.

Using DRUMBEAT to analyze the deactivation MD simulation trajectories of $\beta_2AR$, we uncovered a mechanism of deactivation (Fig. 8A) where the TM7 is pulled out of the G-protein binding site by helix 8 through a transition enabler residue community around the contact $C327^{7.54}$_$F332^{8.50}$; this precedes the inward movement of TM6 movement towards TM3 effector community around $R131^{3.50}$_$E268^{6.30}$, a well-known indicator of the inactive state. This sequential relationship suggests a previously unknown causal mechanism, with the enabler's coordinated motions serving as a precursor to the effector's larger conformational changes.

The DRUMBEAT analysis revealed further similarities and differences between the two dopamine receptors, $D_2R$ and $D_3R$. We found that in $D_2R$, the main mechanism of deactivation initiates at the central region of the receptor with movement of TM7 followed by TM3 and TM6 moving closer together (Fig. 8B). In contrast, the primary deactivation mechanism in $D_3R$ is characterized by an initial allosteric movement of the extracellular region of TM4 and intracellular region of TM3 which enables the movement of TM6 towards TM3 (Fig. 8C). Despite these differences, these receptors share a fundamental reliance on specific contact networks for transitioning between states, illustrating both conserved and subtype-specific features in GPCR deactivation. The distinct deactivation pathway in the $D_3R$ receptor, together with strong enrichment in the "perfectly specific" positions, leads us to hypothesize that $D_3R$ receptors have significantly and concertedly diverged from other dopamine

receptors, suggesting a process of neo-functionalization in the evolution of this receptor subtype.

Analysis of the differences in the TRACS between highly homologous $D_2R$ and $D_3R$ can be harnessed for the design of subtype-specific ligands. As depicted in Fig. 9A, the residue contact $F170^{4.62}$_$F172^{ECL2}$ shows long-range dependencies on $D127^{3.49}$_$Y138^{34.53}$ and $S146^{4.38}$_$G141^{34.56}$ TRACS to enable and effect transition to the inactive state. The TRAC $S146^{4.38}$_$G141^{34.56}$ is involved in the helix to coil transition of the intracellular loop 2 observed during deactivation[25]. This underscores allosteric relationships during transition, such as the information flow between intracellular and extracellular regions that is specific to $D_3R$ and not $D_2R$. Identification of putative small-molecule binding sites using the FindBindSite program[66] (see "Methods") shows (Fig. 9B) that there is a cryptic binding pocket formed near the residues that enable the deactivation transition of $D_3R$. This is a proof of concept for using DRUMBEAT to identify the dynamic elements that differentiate two highly homologous receptors.

Incorporating time steps into a BNM is generally done using dynamic Bayesian networks, which provide insight into the temporal relationships between the features of the system[67]; however, this approach is computationally extremely demanding and lacks precise time anchoring, making it suboptimal for our purpose of dissecting, at the granular level, the sequence of events that drive transitions. DRUMBEAT incorporates time in a notably different, but complementary and equally mathematically rigorous, way that uses the BNM universal graph to identify all possible non-spurious dependencies and then precisely traces where the dependencies manifest alongside a time trajectory. Such granularity not only enhances understanding of GPCR dynamics but also lays the foundation for identifying intervention points in therapeutic contexts.

The implications of this work stem from both the innovative methodology and the results it produced. The DRUMBEAT approach introduces a distinct way to study protein dynamics beyond their stationary states. By using a universal graph to establish probabilistic dependencies across conformational states and a scanning technique for temporal resolution, we identified key residues and events driving transitions. This dual capability provides critical insight into the most challenging regions of dynamic space—the transitions—by pinpointing when and where the protein deviates from equilibrium. Furthermore, the results propose new, collective reaction coordinates for transitions that move beyond traditional single-coordinate models (e.g., distances between two residues) to incorporate coordinated motions among multiple residues. This opens avenues for advanced sampling techniques to model transitions more efficiently. Importantly, DRUMBEAT is a protein system-agnostic framework whose core parameters, such as the contact inclusion cutoff, trajectory sampling scheme, temporal scanning resolution, and TRAC selection criteria, can be tuned to accommodate different study goals and available computational resources without altering the underlying model. This flexibility enables its application well beyond dopamine receptors, including other GPCR classes and non-GPCR proteins such as nanobodies or folding peptides, with the same procedural framework. The ability to adapt DRUMBEAT parameters allows researchers to balance resolution and computational cost while preserving the method's capacity to resolve the temporally ordered events that shape conformational transitions. In addition to generalizability, the reproducibility of results across different GPCRs and the identification of conserved residues within evolutionary families underscore the methodology's robustness. These findings not only shed light on the mechanics of GPCR transitions but also highlight residues critical for receptor family function, paving the way for experimental validation, mutation studies, and therapeutic targeting.

The DRUMBEAT approach presented in this work is a temporally resolved, interpretable ML model designed to scale with expanding multimodal MD datasets, offering granular, mechanistic insights into protein allostery that have long been understudied. Although GPCRs—

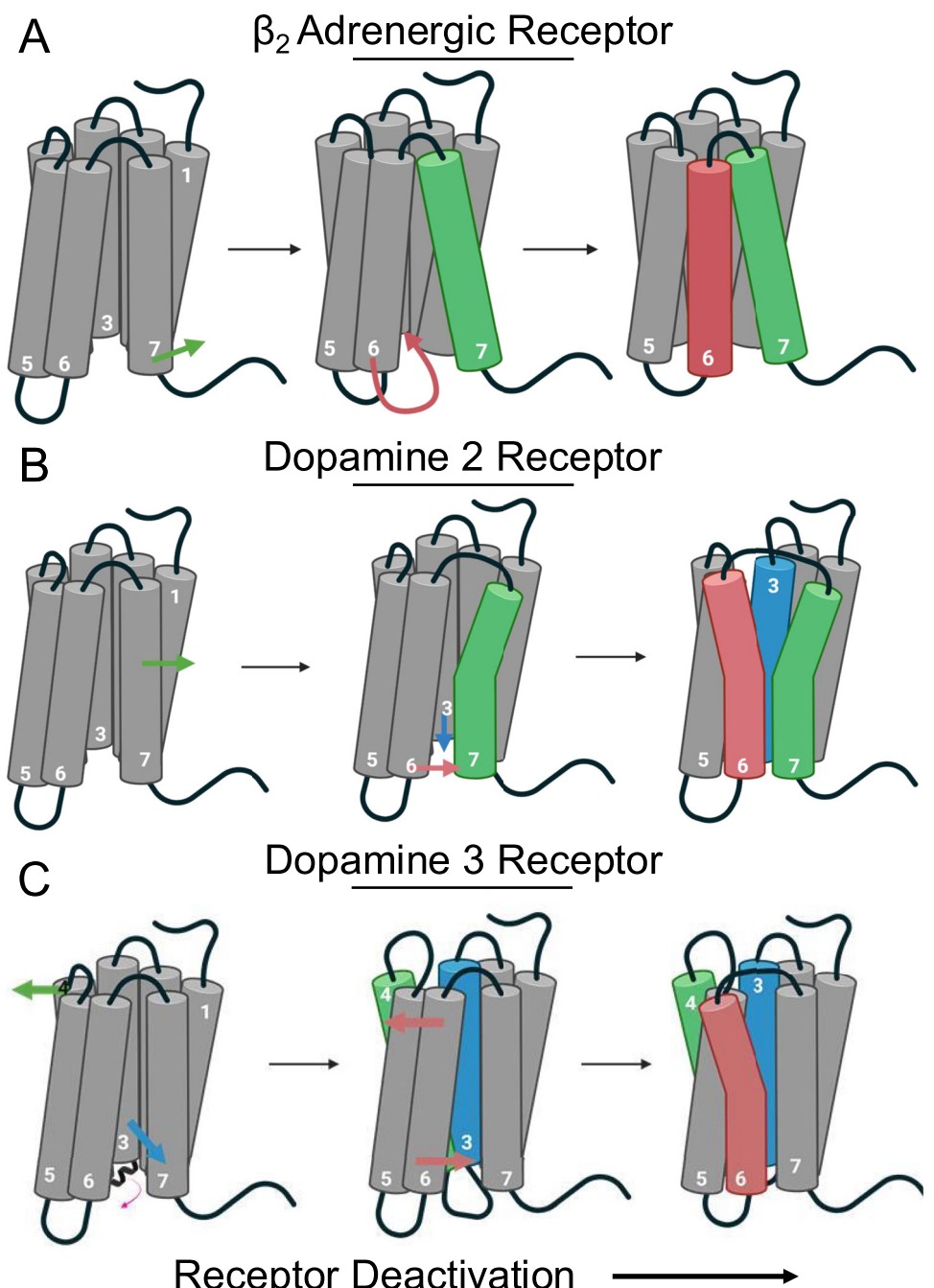

**Fig. 8 | Schematics of the deactivation mechanisms identified by DRUMBEAT.**
**A** Deactivation mechanism for β₂AR is characterized by the outward movement of TM7, followed by the swinging of TM6 towards TM3. Created in BioRender. Mukhaleva, E. (2025) https://BioRender.com/vv8wfke. **B** D₂R deactivation initiates via the movement of TM7 at the central region of the receptor, followed by TM3-TM6 closing. Created in BioRender. Mukhaleva, E. (2025) https://BioRender.com/4t09zyk. **C** D3R deactivation starts with allosteric movement of the extracellular region of TM4 and the intracellular region of TM3 and concludes with movement of TM6. Created in BioRender. Mukhaleva, E. (2025) https://BioRender.com/sxgmojh.

among the most complex allosteric proteins—serve as an ideal model system, the methodology is readily adaptable to other proteins and compounds, providing a temporally resolved analysis that outperforms traditional network models and DL methods in its optimal balance of interpretability, precision, and efficiency. As data continue to grow, the linear computational complexity (given the universal graph) and transparent parametrization and hyperparametrization of this approach ensure that its generalizability will only increase, paving the way for breakthroughs across a wide range of biological systems. Finally, the algorithm is in principle applicable to any contiguous (temporal, spatial)

and sufficiently granular data—adaptation of DRUMBEAT to different such modalities will be a focus of our future studies.

## Methods

### Preparation of protein structures for MD simulations

MD simulations were set up using experimental structures specific to each protein: D₂R (PDB ID: 8IRS)[26] and D₃R (PDB ID: 8IRT)[26]. Agonists and G proteins were excluded from the structures. Missing side chains and loops, involving fewer than five residues, were reconstructed in the 3D models using Schrödinger's Maestro software (Schrödinger Release 2020-1, Schrödinger, LLC, New York, 2020). Local minimization was

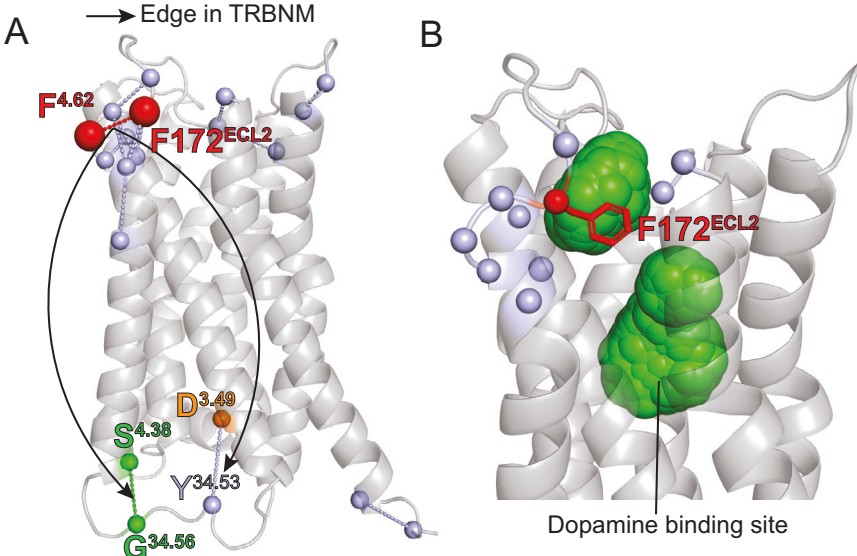

**Fig. 9 | Putative D₃R-specific cryptic binding sites. A** Residue contact in the F170$^{4.62}$_F172$^{ECL2}$ TRAC shows direct allosteric co-dependencies on the intracellular TRACs specific to D₃R identified in the DRUMBEAT analysis. **B** Green spheres are putative small-molecule binding sites identified using the FindBindSite program.

performed for residues within 5 Å of the mutation sites using Macro-Model, while maintaining position restraints on all backbone atoms. Neutral acetyl and methylamide groups were used to cap protein termini, and histidine protonation states were determined using Maestro's protein preparation wizard. Receptors were inserted into an explicit POPC bilayer membrane using the PPM 2.0 functionality of the Orientation of Proteins in Membranes (OPM) tool[68]. The system was then solvated with TIP3P water and 0.15 M NaCl, prepared through CHARMM-GUI[69,70]. Final simulation boxes measured approximately 90 × 90 × 130 Å, and all proteins were modeled using the CHARMM36m force field[71].

### Details of the MD simulations
MD simulations were performed using GROMACS 2022[72] with an integration timestep of 2 fs. The prepared systems underwent an initial energy minimization phase, applying position restraints of 10 kcal/mol Å² to the heavy atoms of proteins, ligands, and lipids where applicable. This was followed by a 1-ns heating phase in the NVT ensemble, gradually increasing the temperature from 0 to 310 K using the Nosé-Hoover thermostat. Subsequent equilibration in the NPT ensemble involved an initial 1-ns run with the same 10 kcal/mol Å² restraints, which were progressively reduced (10, 5, and 1 kcal/mol Å²) over 5-ns intervals. A final 50-ns equilibration phase was conducted without restraints. The resulting configuration from this phase served as the starting structure for thirty-five production simulations, each initialized with distinct random velocities. Pressure was maintained at 1 bar using the Parrinello-Rahman[73] barostat, while nonbonded interactions were calculated with a 12 Å cutoff. Long-range electrostatics were treated using the Particle mesh Ewald method[74], and the LINCS algorithm was employed to constrain all bonds and water molecule angles.

### MD data for β₂-adrenergic receptor
MD trajectories of deactivation of β₂AR were procured from DE Shaw and Associates; these simulations were performed and published previously[25]. An ensemble of trajectories that showed a transition from active to inactive was used for each system. We utilized 14 productive trajectories of condition A (protonated D130)[25].

### Calculating fingerprints of inter-protein pairwise interactions
Pairwise residue contacts between protein residues were analyzed using the Python script library "GetContacts"[75] (https://www.github.com/getcontacts). This tool was employed to identify different types of interactions, including salt bridges (cutoff <4.0 Å), hydrogen bonds (cutoff <3.5 Å with an angle <70°), van der Waals contacts (difference <2 Å), π-stacking interactions (distance <7.0 Å with an angle <30° between aromatic planes), and cation-π interactions (distance <6.0 Å with an angle <60°). The analysis was conducted over the whole trajectory, excluding lipids, water, and ions. Both atom selection groups were aligned with receptor residues. Custom Python scripts were used for one-hot encoding, generating binary contact fingerprints for each simulation frame, where "1" indicates a contact and "0" represents its absence.

### Preprocessing of the MD simulation data
Given $C^{all} = \{c_i\}$, the set of all contacts between two residues in a protein, the following steps were used for feature selection. First, contacts between neighboring residues were excluded from the data. Second, within each trajectory, contacts providing little to no information on other contacts were omitted. Specifically, pairwise MI was computed between contacts in each trajectory. Whichever contacts failed to have a pairwise MI value above 0.01 bits were also removed. This procedure is implemented in BaNDyT software[27]. Finally, the remaining features (contacts) were chosen in the previous steps across all individual trajectories to obtain the set of the most significant contacts $C = \{c_i\}$ with $C \subset C^{all}$.

The universal dataset $D^U = \{c_{ir}\}$ with $c_{ir}$ representing the state of the contact $i$ ($c_{ir} \in \{0, 1\}$) at each frame ($r$) is obtained by (1) concatenating all trajectories or (2) random sampling of frames from the individual trajectories. In general, the first technique is preferred to avoid data loss, but given long trajectories with many frames, concatenation results in a dataset that is too large for the BNM construction in a reasonable timeframe.

In our practice, given that the Bayesian network topology recovery is generally worst-case NP-hard, we found a dataset of roughly 1000 contacts and 15,000 frames to be near the limit of computational feasibility (1 week on a HPCC). For all three receptors, the sampling technique was carried out to provide a single dataset for each system. Since we were interested in specifically studying the transition of the receptor, an enriched sampling was carried out around the transition point for each trajectory to strengthen potentially borderline edges in the resulting BNM structures. Using the interhelical distance as a proxy

for transition, states were randomly sampled (with replacement) from a region of 5000 frames centered at the drop in interhelical distance. The number of states sampled per trajectory was defined by the total number of trajectories for each system and the upper limit of 15,000 frames. For instance, 1200 frames were sampled from each of the 12 trajectories for $\beta_2$AR, giving rise to a total of 14,400 frames. Similarly, 1400 frames were sampled from each $D_2R$ trajectory, and 1200 frames were sampled from each $D_3R$ trajectory, yielding 15,400 and 14,400 frame datasets, respectively. The resulting universal dataset served as an input for the BNM universal graph construction.

## Bayesian network model
For a given set of contacts $C$ and a universal dataset $D^U$ constructed as described above, we used BaNDyT software[27], a specialized implementation of BNOmics[76,77] for MD simulations, to build the Bayesian network for each system. Let $G = (C, \mathcal{E})$ be a Directed Acyclic Graph model with $C$ the set of contact and $\mathcal{E} = \{e_{ij}\}$ the set of edges between contact $i$ and $j$ obtained using BaNDyT, then, the joint probability $\mathcal{P}(C)$ can be factorized as:

$$P(C) = \prod_i p(c_i, |, pa_{c_i})  \quad (1)$$

with $pa_{c_i} \subseteq C$ the set of all parents of $c_i$ in $\mathcal{G}$. Equation (1) states that the joint probability can be represented as the product of the conditional probabilities of each $c_i$ given its parents $pa_{c_i}$.

## Dynamically resolved universal model for Bayesian network tracking
A DRUMBEAT (https://github.com/bandyt-group/drumbeat) was built for each trajectory ($k$) using the universal graph, $\mathcal{G}$. Specifically, given MD simulation data $D^k$ of a trajectory and the universal graph, $\mathcal{G}$, DRUMBEAT uses a sliding window strategy to detect if a specific direct dependency is present in a given temporal interval ($2\delta$) from row in $D^k$. A time-dependent mutual information (MI) serves as a non-parametric measure to quantify the dependency strength between the two nodes connected by an edge in $\mathcal{G}$ within the temporal interval $t - \delta$ to $t + \delta$:

$$MI_t(i, j|\delta) = H(X_{i, t-\delta:t+\delta}) + H(X_{j, t-\delta:t+\delta}) - H(X_{i,j, t-\delta:t+\delta}) \quad (2)$$

An optimization procedure is applied for each edge to determine the optimal window size by maximizing MI across all possible window sizes. Multiple window sizes were used, ranging from 150 to 1050 time steps.

Finally, we define the Time-Resolved Allosteric Community (TRAC) for the contact $i$ as the time-resolved weighted degree given the universal graph $\mathcal{G}$:

$$TRAC_i(t) = \sum_{j \in \Gamma_i} MI_t(i, j|\delta),  \quad (3)$$

with $\Gamma_i$ being the neighbors of the vertex i in $\mathcal{G}$.

## Reporting summary
Further information on research design is available in the Nature Portfolio Reporting Summary linked to this article.

## Data availability
MD simulation trajectories of $\beta_2$AR deactivation are available upon request from the DE Shaw Research Group (https://www.deshawresearch.com/). The wild-type and mutant MD simulation trajectory data in this study have been deposited in the GPCRmd.org database under Dynamics IDs: $D_2R-2376$, $D_3R-2366$, $D_2Rmut-2368$, $D_3Rmut-2367$. The universal networks for $\beta_2$AR, $D_2R$, $D_3R$, and mutants data generated in this study are provided in the Supplementary Information. Source data are provided with this paper and

deposited on Zenodo [https://doi.org/10.5281/zenodo.17478944]. Source data are provided with this paper.

## Code availability
DRUMBEAT code is available on GitHub at https://github.com/bandyt-group/drumbeat and Zenodo [https://doi.org/10.5281/zenodo.17478766][78].

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

## Acknowledgements
This work was funded by grants from the National Institutes of Health R01-GM117923 to N.V., R01-LM013876 to N.V., A.S.R., S.Bh., and S.Br., and R01-LM013138 to A.S.R. The content is solely the responsibility of the authors and does not necessarily represent the official views of the National Institutes of Health. Additional support is acknowledged by the Dr. Susumu Ohno Endowed Chair in Theoretical Biology (held by A.S.R.) and Susumu Ohno Distinguished Investigator Fellowship (to G.G.). This work was supported by NIH grant R35GM156498. S.S. acknowledges support from NIH MIRA award R35-GM126940.

## Author contributions
B.M. developed the methodology, prepared and curated data, implemented and applied the DRUMBEAT software, interpreted the results of the analysis and wrote the first draft of the manuscript. E.M. performed MD simulations of the $D_2R$ and $D_3R$ receptors, prepared contact data and interpreted DRUMBEAT analysis results. GG developed the BNOmics software. B.M., E.M., S.Bh., S.S., N.V., A.S.R., S.Br. conceptualized the study and revised and edited the final version of the manuscript. S.Bh., N.V., A.S.R., S.Br. acquired funding and supervised the study.

## Competing interests
The authors declare no competing interests.
