## [Transparent Peer Review file · Nature Communications]

Temporally Resolved and Interpretable Machine Learning Model of GPCR Conformational Transition

Corresponding Author: Professor Sergio Branciamore

Version 0:

Reviewer comments:

Reviewer #1

(Remarks to the Author)

The manuscript shares some of the ideas of Ref. 56 (Do HN, Wang J, Miao Y. Deep Learning Dynamic Allostery of G-Protein-Coupled Receptors. JACS Au. 2023) and rather weakly fits the Nature Communication scope. However, after significant improvements, including an experimental verification, it could be reconsidered.

Major:

- Bayesian networks can be supervised or unsupervised, specify which case it is. If any assignment of states active/inactive is done, then it is rather supervised.

- Page 4 – it is strange that D2R has different ‘deactivation’ time than D3R. These receptors are close homologs with 56.52 % sequence identity. It suggests that their activation (or ‘deactivation’) scheme is common. D3R is also very close to adrenergic receptors (ca. 40 %), and it is beneficial to add them to the comparison in the manuscript like Authors did, though some other receptors, not that similar to dopamine receptors should also be included. What is more, Rotigotine, the agonist used in both starting structures for MD, is a pan-agonist for all dopamine receptors, including D2R and D3R. If the deactivation mechanisms of these two receptors are indeed different it should be reported or confirmed in time-resolved experimental studies, e.g. differences in signal decay, etc. Authors should provide such confirmation, otherwise it is too hypothetical in its basis. A blind test should be done, confirming that the proposed approach is reliable in detecting residues relevant for receptor deactivation.

- According to Xu, P., Huang, S., Krumm, B.E. et al. Structural genomics of the human dopamine receptor system. Cell Res 33, 604–616 (2023). The hallmark of the activation of dopamine receptors includes residues: 6.48-3.40 (‘rotigotine activates receptors mainly through residue W6.48’). Why were they not included by Authors? Is this microswitch indeed inactive in D2R after 300-600ns? Why this switch hasn’t been detected and included in Fig. 5?

- Distances 3.50-6.34 and 5.58-7.53, although indeed significantly different in the active and inactive state of the receptor, refer to global macroswitches (helix rotation), not microswitches like 6.48. It should be noted by Authors.

- Fig. 8 – are these cryptic allosteric sites in D3R confirmed?

- Fig. S2 – only 12 out of 35 trajectories for D3R were successful (11 out of 31 for D2R), meaning that there was a transition to an inactive state. Could Authors comment on why deactivation in most trajectories was failed? What about the 6.48-3.40 switch in these failed trajectories?

- Why neural networks were not applied to these contact maps (an image composed of pixels) if they are known to outperform Bayesian networks in such tasks? Bayesian networks are usually used for detecting distant associations not for image recognition tasks. Authors should discuss choosing the specific ML algorithm instead of the others. Did Authors test other algorithms? What are the results?

Minor:

- Rather ‘G protein-coupled’ than ‘G-protein coupled’

- GLP1s is not the common abbreviation for incretin receptors

- ‘serotonergic receptors’ – serotonin receptors?

- ‘community of residues’ – change this term

- Provide the full name of residues involved in the activation (residues 3.50-6.34 and 5.58-7.53).

- Fig. 3 – all plots should be shown in detail and higher resolution in the supplementary otherwise it looks like a cartoon figure.

- Methods – why no cholesterol was included in the lipid bilayer? The system preparation seems to be done automatically in Charmm-gui without any modifications.

- Fig. 4D – what is the Connector component in this plot? Could it be added separately? Could Enablers and Effectors

components be split in this plot since they are distinguished from each other?

(Remarks on code availability)

Reviewer #2

(Remarks to the Author)

In this manuscript, the authors present DRUMBEAT, an algorithm for analyzing conformational ensembles in molecular dynamics simulations to identify protein dynamic behavior. DRUMBEAT implements a temporally resolved Bayesian network analysis, extending the group's previously published BaNDyT algorithm for analyzing protein dynamics. The new method enables unbiased identification of specific regions controlling conformational transitions. For validation, the authors first identify known regions involved in β 2-adrenergic receptor deactivation. Subsequently, they analyze dynamics of dopamine D2 and D3 receptors, identifying distinct non-conserved residue communities specific to D3R conformational transitions.

The manuscript is clearly written and the work is conceptually interesting and valuable, both for its current findings and its potential to uncover new aspects of protein dynamics. The interpretability of the method is commendable, and the authors effectively explain their data. This approach provides deeper understanding of conformational transitions in complex ensembles.

The analysis is methodically executed and well-explained. The unbiased nature of the algorithm is particularly valuable, as the authors objectively discuss the features it highlights. However, in my opinion, a more comprehensive analysis of established motifs (e.g., PIF motif) using this algorithm would strengthen the work.

The authors' hypothesis that GPCRs transition between inactive and active states through diverse mechanisms is intriguing but it is based on the comparison of only three receptors without supporting evidence from alternative methods. While the methodological approach is robust and the findings are significant, the limited receptor sampling and lack of support constrains the strength of the conclusions.

Below I provide more detailed comments.

Comments

- lines 333-335: "This transition occurs via TM6 swinging into the G-protein binding pocket towards TM3, ultimately forming a salt bridge between the central contact of the effector TRAC, R131(3.50) and E268(6.30), indicating the transition of receptor into inactive state."

What about the R3.50-D3.49 salt bridge? Is it also re-formed during the transition to the inactive state? This interaction would be important to analyze, as it is more universally conserved in GPCRs than R3.50-E6.30.

- lines 336-337: "Notably, the connector contact between residues I278(6.40) and Y326(7.53), initially broken during the first transition, is reformed in this state, reiterating its structural role in both transitions."

This is an interesting observation, as residue M257(6.40) stabilizes the inactive state of rhodopsin (see Smith, S. O. *Annu. Rev. Biophys.* 52, 301–317 (2023). doi: 10.1146/annurev-biophys-083122-094909). However, residue 6.40 is not mentioned again. Does this residue play any role in D2R or D3R?

- lines 392-394: "We found that in the MD simulations, the deactivation of D2R is faster in time scale by an order of magnitude when compared to β 2AR and D3R.

This is a remarkable observation. Do the authors have an hypothesis for this difference?

- line 399: "The D131(3.49)-T68(2.38) contact at the intracellular region is determined to be a key contact in both D2R and D3R"

Is this contact also important in β 2AR? This would be expected given the role of D3.49 in the salt bridge of the DRY motif.

- lines 400-402: "In the transmembrane region, D80(2.50), S121(3.39) and N422(7.49) residues form two key contacts in D2R that show a hydrogen bond."

Are these contact also important in β 2AR? This comparison would be valuable, as these contacts involve highly conserved residues.

- lines 405-407: "This contact is unique to the D3R and has been shown via mutation studies to play a significant role in ligand binding although it is not located directly in the ligand binding site."

Do the authors have an hypothesis for this allosteric effect?

- lines 417-419: "This most likely corresponds to the outward movement of TM7 that we observed in the β 2AR study and is a preliminary step for deactivation."

Can this be directly verified in the simulations?

- Figure 4B: The authors show G protein selectivity of different receptor subtypes, but I don't see any discussion in the manuscript. It is tempting to speculate that differences in the deactivation/activation mechanism may be related to G protein

engagement and/or selectivity. Perhaps the authors want to explore this point.

- line 141: "The DRUMBEAT Method: The details of DRUMBEAT are given in the Supporting Information."
I did not find the details of the algorithm in the Supporting Information.

Minor

- The abstract emphasizes drug design, but the manuscript focuses on protein dynamics. Drug design represents a potential application not directly explored in this work.

- lines 338-339: "For instance, the salt bridge between R131(3.50) and E268(6.30) is one of the established indicators of rhodopsin deactivation."

Consider referencing Ballesteros et al. (JBC, 2001; doi: 10.1074/jbc.m103747200) for early ionic lock studies. Also note that position 6.30 varies across class A GPCRs; the R3.50-E6.30 salt bridge exists primarily in specific subfamilies (mainly amine receptors and opsins).

- The authors could consider discussing how this approach compares to other recent methods for analyzing conformational transitions in GPCRs, such as Markov state models or deep learning approaches.

(Remarks on code availability)

Reviewer #3

(Remarks to the Author)

This work proposes the ML approach for obtaining and analyzing GPCRs conformational transitions.

The ML model is built in relation to allosteric regulation of GPCRs activity, aiming at "pinpointing residue communities regulating conformational state ensembles". The latter is another way of describing allosteric communication and signalling via dynamics of the whole protein (PMID: 26562539, PMID: 30439587, PMID: 26939022).

The most important aspect of allosteric communication – causality, which should be discussed explaining the link between the structural dynamics and its reflection in proposed here DRUMBEAT method. The causality of signalling can be checked with the help of allosteric signalling and probing maps precalculated for different GPCRs (<https://allomaps.bii.a-star.edu.sg>, PMID: 32392302) – but this is fully to authors' discretion.

It is questionable to understand how well few top-ranking contacts reflect large conformational changes taking place upon receptor's functional regulation. What other network characteristics reflecting above changes authors may think can be used? It is relevant to Enabler-Effector and Connector Contacts, which are characterised by subnetworks – these are seemingly elements that may answer above questions. Is it possible to obtain their quantitative description?

The conservatism of residues involved into function related conformational transitions is a very interesting and important aspect, as well as diversity observed in different subtypes of receptors. It would be interesting to see conservation and variability of signalling's causality, in which these residues are involved.

The method should be applied to additional receptors (not only Dopamin family) to show that it is generic. What would be the "toolset" of parameters for anticipated generalized analysis. Please, discuss it.

(Remarks on code availability)

Version 1:

Reviewer comments:

Reviewer #1

(Remarks to the Author)

The commonly accepted definition of microswitches describes them as residues triggering conformational changes leading to the active-state receptor structure.

In contrast, Authors would like to present 'inverted microswitches' leading to the inactive-state receptor structure. In this sense detection of such 'transition enablers' or 'effector communities', as they call them, in MD trajectories by machine learning is worth considering by Nature Communications. Nevertheless, I still have doubts regarding the terms 'enablers' and 'residue communities'.

What is more, if these 'inverted microswitches' do not overlap with typical microswitches (Table S1) it means that the receptor activation pathway observed as series of conformational changes of the receptor core is intrinsically different comparing the receptor deactivation pathway. Authors should discuss it and add references describing such receptor behavior among various GPCR classes and hypothesize a possible evolutionary explanation. The newly formed inactive state should be reversible. If it is formed in a different pathway, is it indeed the same to the atomic details as before? And, what is more important, do e.g. 180 and 220 ns structures of D2/3 or D3/2 chimeras indeed represent their inactive state or

just a near-inactive transition state?

Fig. 8 – Authors still has not provided any proof about this allosteric site.

What is the percentage of ‘failed’ simulations when this time cholesterol was included in the systems? Did it improve?

(Remarks on code availability)

Reviewer #2

(Remarks to the Author)

In this revised manuscript, the authors have made substantial efforts to address my comments and concerns. I am satisfied with their comprehensive replies and the improvements to the manuscript, particularly the additional experiments and simulations. Overall, the revised manuscript presents a valuable methodological contribution with important mechanistic insights into GPCR conformational transitions.

(Remarks on code availability)

Reviewer #3

(Remarks to the Author)

revision is satisfactory

(Remarks on code availability)

n/a

We thank the reviewers for their thoughtful comments and suggestions that have improved the manuscript. We have addressed all the comments as shown below. The reviewers' comments are in black text, and our replies are in blue font. The text embedded in the revised manuscript is also shown in this document in red text for the reviewers' convenience.

Legend:

Black text – comments/questions from reviewers

Blue text – author comments in response to the reviewers

Red text – text added/changed in the revised manuscript

Grey text – original manuscript text

Reviewer #1 (Remarks to the Author)

General Comments

The manuscript shares some of the ideas of Ref. 56 (Do HN, Wang J, Miao Y. Deep Learning Dynamic Allostery of G-Protein-Coupled Receptors. JACS Au. 2023) and rather weakly fits the Nature Communication scope.

We agree that there is some similarity between Do et al. study and our work; however, our approach and aims are fundamentally distinct. Specifically, this study uses Gaussian Accelerated MD simulations and deep learning model to: (i) classify the states of the receptor when bound to different types of ligands (i.e. agonists, antagonists, positive and negative allosteric modulators) and (ii) identify the important residue contacts that allosterically modulate the receptor activity on several class A and class B GPCRs, by estimating feature (contact) importance via gradient-based attribution. In ML parlance, the latter is post hoc explainability, rather than true, intrinsic interpretability. In our work, we have developed a **Time-Resolved** interpretable Bayesian Network model with the main goal of delineating the residue contacts that enable macrostate transitions in GPCRs (for example it identifies residue contacts that enable deactivation of β_2 -adrenergic receptor and D₂ and D₃ dopamine receptor from an active to an inactive state), with true, dynamically resolved intrinsic interpretability as one of the main objectives. These transition enablers and effector residue communities are not the same as the microswitches and may or may not show changes in conformation when comparing the active and inactive state ensembles.

While models like the one presented in Do et al identify distinct macrostates and microstates from an ensemble, they generally do not specifically identify sequences of time-correlated events or causative residues/contacts that enable and effect transitions. This information is critical for identifying differences between receptor subtypes. Thus, our method offers new mechanistic insights about state transitions with direct relevance to receptor isoform selectivity and the rational design of allosteric drugs --- novelty and breadth that we believe align well with the scope of Nature Communications.

Major Revisions:

1. Bayesian networks can be supervised or unsupervised, specify which case it is. If any assignment of states active/inactive is done, then it is rather supervised.

In this work, Bayesian networks were used as a strictly unsupervised learning model, without any target variable(s) to classify or predict. Moreover, the only a priori knowledge used in our analysis pipeline was the time of deactivation (which corresponded to a decrease in the TM3-TM6 distance). However, this information was used in the sampling stage (before BN modeling) and in the interpretation of the DRUMBEAT output (after BN modeling), but not in BN modeling.

We revised the manuscript in the following sections as shown below:

In Introduction:

In a fully data-driven **and unsupervised learning mode**, we inferred that the community of residues centered around the contact between C327^{7.54} and F332^{8.50} enables the deactivation transition to the inactive state, and the transition is completed by the ionic lock contact between transmembrane helices 3 and 6 (TM3-TM6) R131^{3.50} and E268^{6.30}.

In Discussion:

In this study, we use a novel interpretable machine learning technique (Dynamically Resolved Universal Model for BayEsiAn network Tracking or DRUMBEAT) to identify, **in a fully unsupervised learning mode**, spatially disperse, non-conserved residue communities that coordinate key conformational transitions in GPCRs.

2. Page 4 – it is strange that D2R has different ‘deactivation’ time than D3R. These receptors are close homologs with 56.52 % sequence identity. It suggests that their activation (or ‘deactivation’) scheme is common. D3R is also very close to adrenergic receptors (ca. 40 %), and it is beneficial to add them to the comparison in the manuscript like Authors did, though some other receptors, not that similar to dopamine receptors should also be included. What is more, Rotigotine, the agonist used in both starting structures for MD, is a pan-agonist for all dopamine receptors, including D2R and D3R.

If the deactivation mechanisms of these two receptors are indeed different it should be reported or confirmed in time-resolved experimental studies, e.g. differences in signal decay, etc. **Authors should provide such confirmation**, otherwise it is too hypothetical in its basis.

We thank the reviewer for this suggestion. The short time scales within which the transition enabler and effector residue communities enable the conformational transitions (~ 250 ns), combined with the inherent transient nature of the transition states, make them inaccessible to testing with established experimental techniques such as time-resolved FRET. Additionally,

precise labeling of the residues that are buried in the TM region can be inaccessible in membrane-bound proteins. Nonetheless, we have taken the reviewer's suggestion and performed MD simulation studies on the predicted TRAC mutants that convert D₂ to be D₃-like and vice versa and calculated the transition times. The results are described in the reply to the next query of the reviewer. We also propose that these enabler and effector residue communities can guide the selection of binding site cavities to focus virtual ligand screening efforts. Future efforts of our team will focus on demonstrating the utility of putative binding sites located near the TRBNM-identified transition state sites in allosteric modulator discovery.

3. A blind test should be done, confirming that the proposed approach is reliable in detecting residues relevant for receptor deactivation.

This is an excellent suggestion. As described in the answer to the previous query, given the inaccessibility of detecting the conformational transitions using experimental approaches, we crafted a blind test for validating the predictions of the transition effector TRACS communities. Specifically, we have generated a chimeric D_{2/3}R that substitutes our TRACs from D₃R for the amino acids in D₂R and vice versa. We conducted 25 independent deactivation MD simulations for each chimera.

As predicted by our model, the D_{2/3}R chimera had a markedly increased deactivation time (median 220 ns vs. 70 ns in wild-type D₂R), while the D_{3/2}R chimera deactivated more quickly (median 180 ns vs. 330 ns in wild-type D₃R). These results confirm that the identified TRACs are critical determinants of deactivation dynamics and directly modulate receptor conformational transitions.

The following figure and corresponding text were added to a new section in the manuscript.

We have added a new subsection after the section titled “Do the transition enabler and effector residues differ among Dopamine Receptor D₂R and D₃R subtypes?”

Swapping Time-Resolved Allosteric Contacts Confirms the Predicted Determinants of Receptor Deactivation Dynamics

To evaluate whether the TRACs we identified using TRBNM for D₂R and D₃R represent critical determinants of the deactivation transition and whether altering them can modify the transition patterns, we performed reciprocal TRAC-swapping mutations between the two receptors at positions where the TRACs differ (Fig. 7A). Specifically, we mutated residues F172/L174 in ECL2 (TRAC F4.62_F172 in D₃R), position 5.66 (TRAC R5.66_Y5.62 in D₂R) by introducing an R→K mutation in D₂R and a K→R mutation in D₃R, and position 34.56 in ICL2 (TRAC G34.56_S4.38 in D₃R) by inserting a residue in D₂R and deleting it in D₃R (Fig. 7B, Table 1). We conducted 25 deactivation MD simulations for each chimera, obtaining 19 successful deactivation trajectories for the D_{2/3}R mutant and 14 for the D_{3/2}R mutant. In both cases, the time required for deactivation differed significantly from their respective wild-type receptors (Fig.

7C). The average deactivation time for the D_{2/3}R chimera increased from 70 ns to 220 ns, whereas for the D_{3/2}R chimera, it decreased from 330 ns to 180 ns.

Table 1. Swapping mutation between D₂R and D₃R.

TRAC	Mutation in D₃R	Mutation in D₂R
F4.62_F172 (D ₃ R)	F172L	L174F
G34.56_S4.38 (D ₃ R)	G141 deletion	G144 insertion
R5.66_Y5.62 (D ₂ R)	K216R	R217K

Figure 7. (A) Alignment of residues in D₂ and D₃R TRACs as identified by DRUMBEAT. Red arrows show the positions where mutagenesis was done. (B) Locations of mutations in the receptor. (C) Data for deactivation times for wild-type and mutant D₂R/D₃R.

To quantitatively assess these differences, we performed Mann-Whitney U tests (Table 2). The results confirmed significant differences between wild-type receptors and their respective mutants, with U statistics of 2 ($p < 0.0001$) for D₂R vs D_{2/3}R and 41 ($p = 0.0283$) for D_{3/2}R vs D₃R. Interestingly, not only did the mutations alter the deactivation time scales, but the mutants also behaved in a manner opposite to their respective wild-type receptors. Specifically, the D_{2/3}R chimera exhibited a deactivation time distribution more similar to D₃R than to D₂R ($U = 65.5$ vs D₃R and $U = 2$ vs D₂R), whereas the D_{3/2}R chimera behaved closer to D₂R than D₃R ($U = 30.5$ vs D₂R and $U = 41$ vs D₃R). These findings indicate that the correct combination of TRACs is

required to ensure efficient progression through the deactivation pathway and that swapping TRACs reverses the receptor's deactivation characteristics.

Table 2. Summary of computed statistics for swapping mutation study.

Comparison (vs groups)		Mann-Whitney Test	
		U statistics	p-value
D ₂ R	D _{3/2} R	30.5	0.0114
D ₂ R	D _{2/3} R	2	0
D ₂ R	D ₃ R	9	0.0005
D _{3/2} R	D _{2/3} R	103	0.2793
D _{3/2} R	D ₃ R	41	0.0283
D _{2/3} R	D ₃ R	65.5	0.0498

4. According to Xu, P., Huang, S., Krumm, B.E. et al. Structural genomics of the human dopamine receptor system. *Cell Res* 33, 604–616 (2023). The hallmark of the activation of dopamine receptors includes residues: 6.48-3.40 ('rotigotine activates receptors mainly through residue W6.48').

Why were they not included by Authors? Is this microswitch indeed inactive in D2R after 300-600ns? Why this switch hasn't been detected and included in Fig. 5?

Sorry about the confusion between TRACS (Temporally Resolved Allosteric Communities) and microswitches. The study cited by the reviewer covers solving the structures of D₁ to D₅ dopamine receptors and comparison of these structures, elucidating the structural differences between the active and inactive states of the receptors. The residues that show distinct conformational changes on comparison of the static structures of the active and inactive states of GPCRs are called microswitches. What we identify and delineate in this study are the residues that enable transitions between the active and inactive states, and not necessarily the microswitches that show conformational change in the two end states. Such "enabler" residues **need not** be microswitch residues. These TRACS momentarily show changes in their co-dependencies in the dynamics to enable transitions. However, they need not show a conformational change when comparing active state to inactive state structures. We compared the inter-residue pairwise distances of the TRACS (shown in Figure 5) in the active and inactive states of D₂R and D₃R. As shown in Table S2 below, they do not show a significant change in the inter-residue distances.

In light of this comment, we decided to add a table to the Supporting Information (Table S1) of the manuscript that summarizes the TRACS identified by DRUMBEAT for each GPCR system, along with information on their known roles as microswitches. We have also added a table to the Supporting Information (Table S2) showing the inter-residue distances for the TRACS in β_2 AR, D₂R, and D₃R in their inactive and active states. As seen in this table, some of the TRACs that coincide with the microswitches show inter-residue distance changes, and others do not.

Added Table:

Table S1: Summary of TRACS determined by DRUMBEAT method for each receptor. A brief description of the functional relevance for each TRAC is also provided. Each TRAC is determined to be either a known microswitch or not, highlighting DRUMBEAT's ability to recapitulate known microswitches while also providing insight into new contacts that facilitate transition.

Protein	DRUMBEAT TRAC residue contacts	Receptor Function Relevance	Known Microswitch
β2AR	R3.50_E6.30	Ionic lock; stabilizes inactive state in many class A GPCRs. [1]	Yes
	C7.54_F8.50	Part of NPxxY/F motif; essential for G-protein coupling. [2]	Yes
	C7.54_V1.53	TM1–TM7 contact; aids structural stability during transitions.	No
	R8.51_C7.54	Supports helix 8-TM7 interface; involved in signaling specificity.	No
	F8.50_Y7.53	Aromatic cluster; key in conformational switch (NPxxY region). [3]	Yes
	S8.47_Y7.53	Modifies NPxxY switch dynamics; stabilizes helix 8 rearrangement.	No
	L3.43_Y7.53	Facilitates TM3-TM7 cross-talk during receptor transitions.	No
D2R	D2.50_S3.39	Core sodium pocket; allosteric regulation in class A GPCRs. [4]	Yes
	N7.49_D2.50	Links NPxxY and sodium pocket; disrupted upon deactivation. [4]	Yes
	N7.45_L7.41	TM7 stability; may support NPxxY function (not a classic switch).	No
	D3.49_T2.38	DRY motif–TM2 link; stabilizes inactive state (subtype specific).	No
	R5.66_Y5.62	TM5 interaction; may aid in cytoplasmic region stability.	No
D3R	D3.49_T2.38	DRY motif–TM2 link; stabilizes inactive state (subtype specific).	No
	C6.47_F6.51	TM6 toggle switch region; central in activation. [5]	Yes
	M3.56_Y3.51	TM3 contacts; support DRY motif packing and transitions.	No
	F4.62_F172	TM4/loop aromatic contact; may influence subtype signaling.	No
	G34.56_S4.38	TM4 local contact; role not classically recognized.	No

(Table references to be added to revised manuscript according to journal formatting: 1. Latorraca et al 2017 <https://pubs.acs.org/doi/10.1021/acs.chemrev.6b00177>; 2. Fritze et al 2003 <https://www.pnas.org/doi/full/10.1073/pnas.0435715100>; 3. Zhou et al 2019 <https://doi.org/10.7554/eLife.50279>; 4. Katritch et al 2014 <https://doi.org/10.1016/j.tibs.2014.03.002>; 5. Shi et al 2002 <https://doi.org/10.1074/jbc.M206801200>)

Table S2: Inter-residue distance changes in the active and inactive state structures of the TRAC residues for β_2 AR, D₂R and D₃R

TRACs in D ₂ R (GPCRdb residue numbering)	TRAC in D ₂ R	Smallest distance (Å) in active structure 8IRS	Shortest distance (Å) in inactive structure 6LUQ
N7.45_L7.41	N418_L414	3.7	6.8
N7.49_D2.50 (a part of extended sodium binding site)	N422_D80	2.9	9.1
D2.50_S3.39	D80_S121	3.3	3.5
R5.66_Y5.62	R217_Y213	3.2	2.9
D3.49_T2.38	D131_T68	3.3	2.6

TRACs in D ₃ R (GPCRdb residue numbering)	TRAC in D ₃ R	Shortest distance (Å) in active structure 8IRT	Shortest distance (Å) in inactive structure 3PBL
C6.47_F6.51	C341_F345	2.8	4.8
M3.56_Y3.51	M134_Y129	4.2	3.9
F4.62_F172(ECL2)	F170_F172	2.9	3.6
G34.56_S4.38	G141_S146	4.0	3.2
D3.49_T2.38	D127_T63	4.5	3.2

5. Distances 3.50-6.34 and 5.58-7.53, although indeed significantly different in the active and inactive state of the receptor, refer to global macroswitches (helix rotation), not microswitches like 6.48. It should be noted by Authors.

We appreciate the reviewer pointing this out; we have added the following to the main text of the manuscript for clarification.

The sentence in the sub-section titled “*Application of DRUMBEAT to deactivation of β_2 -adrenergic receptor*”

“The contact R131^{3.50}_E268^{6.30} was already previously shown to be a critical "micro-switch" that reduces in distance upon deactivation “

has been changed to: “The contact R131^{3.50}_E268^{6.30} was previously shown to be a critical structural element or a macroswitch (Latorraca at al. 2017

<https://pubs.acs.org/doi/10.1021/acs.chemrev.6b00177>) that undergoes significant conformational change upon activation.”

6. Fig. 8- Are these cryptic allosteric sites in D₃R confirmed?

The small molecule binding site is a predicted binding site located near the residue communities (TRACs) that enable transitions in D₃R and not in D₂R. Hence, we hypothesize that a small molecule in this binding site will be specific to D₃R. Our future work involves using this binding site to identify small molecules that are specific to D₃R.

7. Fig. S2 – only 12 out of 35 trajectories for D₃R were successful (11 out of 31 for D₂R), meaning that there was a transition to an inactive state. Could Authors comment on why deactivation in most trajectories was failed? What about the 6.48-3.40 switch in these failed trajectories?

We thank the reviewer for this comment. The starting velocities of the atoms are chosen using a random seed from a Boltzmann distribution of instantaneous atom velocities at a given temperature. Some of these velocities may not lead to deactivation of the receptor, depending on the conformational sampling afforded by the simulation time scales, starting velocities, and forcefield trapping the protein system in a local minimum. Regarding the inter-residue contact 6.48-3.40, this contact is typically present in active states and absent in the inactive state structures of D₃R and D₂R (differences in distance between residues 6.48 and 3.40 in active and inactive crystal structures are 4.7 Å and 4.3 Å in D₂R and D₃R, respectively). We calculated the C α -C α distance between these residues across all D₂R and D₃R trajectories. We observed no significant difference in this distance between successful and failed deactivation trajectories - the average distance remained consistently within the 8–9 Å range regardless of whether the receptor transitioned to an inactive state (successful) or (failed). Furthermore, the van der Waals contacts between residues 6.48 and 3.40 were maintained for nearly 100% of the simulation time in both successful and failed trajectories.

8. Why neural networks were not applied to these contact maps (an image composed of pixels) if they are known to outperform Bayesian networks in such tasks? Bayesian networks are usually used for detecting distant associations not for image recognition tasks. Authors should discuss choosing the specific ML algorithm instead of the others. Did Authors test other algorithms? What are the results?

Among various machine learning (ML) and deep learning (DL) models considered by us, Bayesian networks (BNs) achieve the best balance of scalability, feature filtering, and, most importantly, interpretability. Specifically, we have chosen BNs as the primary network-centric method for several reasons: (1) intrinsic mechanistic interpretability, (2) ability to filter out induced, or spurious, dependencies, (3) ability to handle directional (e.g., temporal) causality, which is essential for the time-resolved analyses, (4) appropriate dimensionality (hundreds of residues/contacts X long-scale MD simulation trajectories of varying granularity) that enables scalability and computational efficiency, (5) ability to directly compare and contrast similar but different networks (e.g., between different homologous proteins or before vs. after transition), and (6) ability to extract and utilize local- and global-context network properties (e.g., weighted degree of key nodes).

Importantly, we have deliberately chosen naturally interpretable ML models over various DL approaches. While the latter are undoubtedly promising in the context of MD simulations' generation, sampling, and extension (D'Hondt et al. 2025 <https://doi.org/10.1186/s13321-025-00985-7>, Lewis et al 2025 <https://www.science.org/doi/10.1126/science.adv9817>), their applicability to the MD data analysis is limited to dimensionality reduction/feature extraction (Zhang et al 2025 <https://doi.org/10.1093/bioinformatics/btaf321>, Mustali et al 2023 <https://pubs.rsc.org/en/content/articlelanding/2023/ra/d3ra06375e>) or supervised learning (classification) with at most post-hoc explainability, such as one-way feature importance ranking (Plante et al 2019 <https://doi.org/10.3390/MOLECULES24112097>, Plante et al 2021 <https://doi.org/10.3390/MOLECULES26103059>, Do et al 2022 <https://doi.org/10.1021/acs.jctc.1c01055>, Moraes doe Santos et al 2024 <https://doi.org/10.1038/s41598-024-72842-w>). This was not our goal, and it contrasts sharply with our BN approach --- although it is as fully data-driven and as sensitive to rare events and nonlinearity as DL, its deliverables are not the classification accuracy and/or feature importance ranking, but rather the intrinsically interpretable multi-way interactions represented as dynamic network structures (directed acyclic graphs, DAGs).

It remains to note that (i) BNs detect all associations, not only the distant ones, and (ii) there is no need to encode the MD data as an image, because all data types can be used as direct BN inputs (as opposed to a CNN). We did not directly compare DL to BN, because the ML activities are fundamentally different and largely incommensurate --- classification/feature importance ranking vs. dynamic network modeling, respectively.

Minor Revisions:

1. Rather 'G protein-coupled' than 'G-protein coupled'

Thank you to the reviewer for pointing this out. We have corrected it in the manuscript.

2. GLP1s is not the common abbreviation for incretin receptors

Thank you to the reviewer for pointing this out. We have corrected it in the manuscript.

3. -'serotonergic receptors' – serotonin receptors?

We have corrected it in the manuscript

4. - 'community of residues' – change this term

Thank you for the suggestion. After careful consideration, we have decided to retain the term "community of residues," as we believe it appropriately reflects the network-based context in which it is used.

5. - Provide the full name of residues involved in the activation (residues 3.50-6.34 and 5.58-7.53)

We included the residue names.

The paragraph at the beginning of the *Results* Section was changed:

The distance between residues **R3.50-T6.34** and **Y5.58-Y7.53** (using GPCR data base residue numbering system) were calculated for each trajectory in every GPCR system and plotted as 2D density plots shown in Fig. S1 and S2. Trajectories 1-11 for D₂R and 1-12 for D₃R did show a transition from active to inactive state as indicated by the decrease in the distance between the residues **R3.50 and T6.34** on transmembrane helices 3 and 6 (TM3 and TM6) and simultaneous increase in the distance between **Y5.58 and Y7.53** in TM5-TM7.

6. - Fig. 3 – all plots should be shown in detail and higher resolution in the supplementary otherwise it looks like a cartoon figure.

Thank you for the comment. We have added high-definition .svg for all Figures in the manuscript and SI.

7. - Methods – why no cholesterol was included in the lipid bilayer? The system preparation seems to be done automatically in Charmm-gui without any modifications.

The original β_2 AR simulations by Dror et al. did not include cholesterol, so we matched these conditions in our D₂R and D₃R simulations to ensure a consistent comparison across systems. However, in response to your suggestion, we have included cholesterol in the lipid bilayer for the D₂R and D₃R mutation simulations to better reflect physiological membrane composition.

8. - Fig. 4D – what is the Connector component in this plot? Could it be added separately? Could Enablers and Effectors components be split in this plot since they are distinguished from each other?

We thank the reviewer for this question. In this plot, we considered the conservation score of specific sets of residues identified from the DRUMBEAT analysis at multiple levels. We show that the connector contact residues are also scored highly in comparison to the set of all residues. The comparison between enabler and effector (red) and enabler and effector including connectors (green) shows that most of the conservation is for the residues in the enabler and effector residues, and the connector residues are also conserved but not as strongly.

Reviewer #2 (Remarks to the Author)

General Comments

In this manuscript, the authors present DRUMBEAT, an algorithm for analyzing conformational ensembles in molecular dynamics simulations to identify protein dynamic behavior.

DRUMBEAT implements a temporally resolved Bayesian network analysis, extending the group's previously published BaNDyT algorithm for analyzing protein dynamics. The new method enables unbiased identification of specific regions controlling conformational transitions. For validation, the authors first identify known regions involved in β 2-adrenergic receptor deactivation. Subsequently, they analyze dynamics of dopamine D2 and D3 receptors, identifying distinct non-conserved residue communities specific to D3R conformational transitions.

The manuscript is clearly written and the work is conceptually interesting and valuable, both for its current findings and its potential to uncover new aspects of protein dynamics. The interpretability of the method is commendable, and the authors effectively explain their data. This approach provides deeper understanding of conformational transitions in complex ensembles.

We thank the reviewer for the positive comments on the significance of our study.

The analysis is methodically executed and well-explained. The unbiased nature of the algorithm is particularly valuable, as the authors objectively discuss the features it highlights. However, in my opinion, a more comprehensive analysis of established motifs (e.g., PIF motif) using this algorithm would strengthen the work.

We thank the reviewer for the comment. Sorry about the confusion between Temporally Resolved Allosteric Communities (TRACs) identified in this work, which are transition enablers and effector residue pairs, and the microswitches, such as the PIF motif. These TRACS momentarily show changes in their co-dependencies in the dynamics to enable transitions. However, they may or may not show a conformational change when comparing active state to inactive state structures. Taking the reviewers' suggestion, we compared the inter-residue pairwise distances of the TRACS (shown in Figure 5) in the active and inactive states of D₂R and D₃R. As shown in Table S2 (below; added to Supplementary Information), a subset of the TRACS shows a large conformational change upon deactivation, while others are comparatively unchanged and instead may affect conformational changes in other residue pairs. In addition, as suggested by the reviewer, we have also done a comprehensive analysis of the TRACS and their relationship to microswitches and added a summary Table to the Supporting Information showing if the TRACS residues are the microswitch residues or not (see Table S1 below).

Added Tables:

Table S1: Summary of TRACS determined by DRUMBEAT method for each receptor. A brief description of the functional relevance for each TRAC is also provided. Each TRAC is determined to be either a known microswitch or not, highlighting DRUMBEAT's ability to recapitulate known microswitches while also providing insight into new contacts that facilitate transition.

Protein	DRUMBEAT TRAC residue contacts	Receptor Function Relevance	Known Microswitch
β 2AR	R3.50_E6.30	Ionic lock; stabilizes inactive state in many class A GPCRs.[1]	Yes
	C7.54_F8.50	Part of NPxxY/F motif; essential for G-protein coupling.[2]	Yes
	C7.54_V1.53	TM1–TM7 contact; aids structural stability during transitions.	No
	R8.51_C7.54	Supports helix 8-TM7 interface; involved in signaling specificity.	No
	F8.50_Y7.53	Aromatic cluster; key in conformational switch (NPxxY region).[3]	Yes
	S8.47_Y7.53	Modifies NPxxY switch dynamics; stabilizes helix 8 rearrangement.	No
	L3.43_Y7.53	Facilitates TM3-TM7 cross-talk during receptor transitions.	No
D2R	D2.50_S3.39	Core sodium pocket; allosteric regulation in class A GPCRs.[4]	Yes
	N7.49_D2.50	Links NPxxY and sodium pocket; disrupted upon deactivation.[4]	Yes
	N7.45_L7.41	TM7 stability; may support NPxxY function (not a classic switch).	No
	D3.49_T2.38	DRY motif–TM2 link; stabilizes inactive state (subtype specific).	No
	R5.66_Y5.62	TM5 interaction; may aid in cytoplasmic region stability.	No
D3R	D3.49_T2.38	DRY motif–TM2 link; stabilizes inactive state (subtype specific).	No
	C6.47_F6.51	TM6 toggle switch region; central in activation.[5]	Yes
	M3.56_Y3.51	TM3 contacts; support DRY motif packing and transitions.	No
	F4.62_F172	TM4/loop aromatic contact; may influence subtype signaling.	No
	G34.56_S4.38	TM4 local contact; role not classically recognized.	No

(Table references to be added to revised manuscript according to journal formatting: 1. Latorraca et al 2017 <https://pubs.acs.org/doi/10.1021/acs.chemrev.6b00177>; 2. Fritze et al 2003 <https://www.pnas.org/doi/full/10.1073/pnas.0435715100>; 3. Zhou et al 2019 <https://doi.org/10.7554/eLife.50279>; 4. Katritch et al 2014 <https://doi.org/10.1016/j.tibs.2014.03.002>; 5. Shi et al 2002 <https://doi.org/10.1074/jbc.M206801200>)

Table S2: Inter-residue distance changes in the active and inactive state structures of the TRAC residues for β 2AR, D2R and D3R

TRACs in D ₂ R (GPCRdb residue numbering)	TRAC in D ₂ R	Smallest distance (Å) in active structure 8IRS	Shortest distance (Å) in inactive structure 6LUQ
N7.45_L7.41	N418_L414	3.7	6.8
N7.49_D2.50 (a part of extended sodium binding site)	N422_D80	2.9	9.1
D2.50_S3.39	D80_S121	3.3	3.5
R5.66_Y5.62	R217_Y213	3.2	2.9
D3.49_T2.38	D131_T68	3.3	2.6

TRACs in D ₃ R (GPCRdb residue numbering)	TRAC in D ₃ R	Shortest distance (Å) in active structure 8IRT	Shortest distance (Å) in inactive structure 3PBL
C6.47_F6.51	C341_F345	2.8	4.8
M3.56_Y3.51	M134_Y129	4.2	3.9
F4.62_F172(ECL2)	F170_F172	2.9	3.6
G34.56_S4.38	G141_S146	4.0	3.2
D3.49_T2.38	D127_T63	4.5	3.2

The authors' hypothesis that GPCRs transition between inactive and active states through diverse mechanisms is intriguing but it is based on the comparison of only three receptors without supporting evidence from alternative methods. While the methodological approach is robust and the findings are significant, the limited receptor sampling and lack of support constrains the strength of the conclusions.

We are grateful to the reviewer for the thoughtful notes on the robustness and significance of the study. In regard to the limited receptor sampling and lack of support, we aimed to evaluate whether the TRACs we identified using TRBNM for D₂R and D₃R represent critical determinants of the deactivation transition and whether altering them can modify the transition patterns by performing reciprocal TRAC-swapping mutations between the two receptors at positions where the TRACs differ (Fig. 7A). Specifically, we mutated residues F172/L174 in ECL2 (TRAC F4.62_F172 in D₃R), position 5.66 (TRAC R5.66_Y5.62 in D₂R) by introducing an R→K mutation in D₂R and a K→R mutation in D₃R, and position 34.56 in ICL2 (TRAC G34.56_S4.38 in D₃R) by inserting a residue in D₂R and deleting it in D₃R (Fig. 7B, Table 1). We conducted 25 deactivation MD simulations for each chimera, obtaining 19 successful deactivation trajectories for the D_{2/3}R mutant and 14 for the D_{3/2}R mutant. In both cases, the time required for deactivation differed significantly from their respective wild-type receptors (Fig. 7C). The average deactivation time for the D_{2/3}R chimera increased from 70 ns to 220 ns,

whereas for the D_{3/2}R chimera, it decreased from 330 ns to 180 ns. In light of these findings, we have added the following section to the revised manuscript.

We have added a new subsection after the section titled “Do the transition enabler and effector residues differ among Dopamine Receptor D₂R and D₃R subtypes?”

Swapping Time-Resolved Allosteric Contacts Confirms the Predicted Determinants of Receptor Deactivation Dynamics

To evaluate whether the TRACs we identified using TRBNM for D₂R and D₃R represent critical determinants of the deactivation transition and whether altering them can modify the transition patterns, we performed reciprocal TRAC-swapping mutations between the two receptors at positions where the TRACs differ (Fig. 7A). Specifically, we mutated residues F172/L174 in ECL2 (TRAC F4.62_F172 in D₃R), position 5.66 (TRAC R5.66_Y5.62 in D₂R) by introducing an R→K mutation in D₂R and a K→R mutation in D₃R, and position 34.56 in ICL2 (TRAC G34.56_S4.38 in D₃R) by inserting a residue in D₂R and deleting it in D₃R (Fig. 7B, Table 1). We conducted 25 deactivation MD simulations for each chimera, obtaining 19 successful deactivation trajectories for the D_{2/3}R mutant and 14 for the D_{3/2}R mutant. In both cases, the time required for deactivation differed significantly from their respective wild-type receptors (Fig. 7C). The average deactivation time for the D_{2/3}R chimera increased from 70 ns to 220 ns, whereas for the D_{3/2}R chimera, it decreased from 330 ns to 180 ns.

Table 1. Swapping mutation between D₂R and D₃R.

TRAC	Mutation in D₃R	Mutation in D₂R
F4.62_F172 (D ₃ R)	F172L	L174F
G34.56_S4.38 (D ₃ R)	G141 deletion	G144 insertion
R5.66_Y5.62 (D ₂ R)	K216R	R217K

Figure 7. (A) Alignment of residues in D₂ and D₃R TRACs as identified by DRUMBEAT. Red arrows show the positions where mutagenesis was done. (B) Locations of mutations in the receptor. (C) Data for deactivation times for wild-type and mutant D₂R/D₃R.

To quantitatively assess these differences, we performed Mann-Whitney U tests (Table 2). The results confirmed significant differences between wild-type receptors and their respective mutants, with U statistics of 2 ($p < 0.0001$) for D₂R vs D_{2/3}R and 41 ($p = 0.0283$) for D_{3/2}R vs D₃R. Interestingly, not only did the mutations alter the deactivation time scales, but the mutants also behaved in a manner opposite to their respective wild-type receptors. Specifically, the D_{2/3}R chimera exhibited a deactivation time distribution more similar to D₃R than to D₂R ($U = 65.5$ vs D₃R and $U = 2$ vs D₂R), whereas the D_{3/2}R chimera behaved closer to D₂R than D₃R ($U = 30.5$ vs D₂R and $U = 41$ vs D₃R). These findings indicate that the correct combination of TRACs is

required to ensure efficient progression through the deactivation pathway and that swapping TRACs reverses the receptor's deactivation characteristics.

Table 2. Summary of computed statistics for swapping mutation study.

Comparison (vs groups)		Mann-Whitney Test	
		U statistics	p-value
D ₂ R	D _{3/2} R	30.5	0.0114
D ₂ R	D _{2/3} R	2	0
D ₂ R	D ₃ R	9	0.0005
D _{3/2} R	D _{2/3} R	103	0.2793
D _{3/2} R	D ₃ R	41	0.0283
D _{2/3} R	D ₃ R	65.5	0.0498

Comments

1. lines 333-335: “This transition occurs via TM6 swinging into the G-protein binding pocket towards TM3, ultimately forming a salt bridge between the central contact of the effector TRAC, R131(3.50) and E268(6.30), indicating the transition of receptor into inactive state.”

What about the R3.50-D3.49 salt bridge? Is it also re-formed during the transition to the inactive state? This interaction would be important to analyze, as it is more universally conserved in GPCRs than R3.50-E6.30.

The reviewer is right that the R3.50-D3.49 has been observed in the inactive state crystal structures of class A GPCRs. However, it was excluded from our analysis due to our preprocessing step, which filters out contacts between neighboring residues to reduce noise and computational load. As a result, this specific interaction was not evaluated by our model.

However, in response to the reviewer’s comment, we analyzed the β_2 AR trajectories and found that the R131(3.50)–D130(3.49) salt bridge does not form at all in any of the β_2 AR deactivation simulations. The distance between these residues, measured between side-chain atoms, decreases upon deactivation (from ~ 10 Å to ~ 7 Å) mediated by the movement of TM6, but not enough to form a contact. Thus, in the case of these β_2 AR simulations, the intracellular salt bridge is not formed during the deactivation simulations in contrast to the R131(3.50)–E268(6.30) salt bridge which forms consistently across all trajectories.

The manuscript has been revised as follows:

This transition occurs via TM6 swinging into the G-protein binding pocket towards TM3, ultimately forming a salt bridge between the central contact of the effector TRAC, R131(3.50) and E268(6.30), indicating the transition of receptor into inactive state. **We note here that the residue R3.50 could also form a salt bridge with its neighbor D3.49 as seen in the crystal structures of the inactive state of some class A GPCRs (Zhou et al 2020 <https://doi.org/10.7554/eLife.50279>).** We examined this contact in the β_2 AR trajectories and

found it does not consistently form upon deactivation, in line with previous reports of significant basal activity for β_2 AR leading to the lack of formation of the R3.50_D3.49 salt bridge in the inactive state [Rovati et al 2017 <https://pubmed.ncbi.nlm.nih.gov/28347873/>]

2. lines 336-337: “Notably, the connector contact between residues I278(6.40) and Y326(7.53), initially broken during the first transition, is reformed in this state, reiterating its structural role in both transitions.”

This is an interesting observation, as residue M257(6.40) stabilizes the inactive state of rhodopsin (see Smith, S. O. Annu. Rev. Biophys. 52, 301–317 (2023). doi: 10.1146/annurev-iophys-083122-094909). However, residue 6.40 is not mentioned again. Does this residue play any role in D2R or D3R?

We thank the reviewer for the insightful comment and for highlighting the functional relevance of residue 6.40 in rhodopsin and its similarity to β_2 AR. We examined the connector contacts in D2R and D3R to assess whether position 6.40 or adjacent TM6 residues play a similar role.

In D2R, several TM6 contacts were identified among connector residues, including M6.36_Y7.53, W6.48_V6.43, and Q6.28_K6.32. In D3R, we observed N6.58_H6.60 in TM6, which may also contribute to intramolecular communication. However, we did not detect contacts involving position 6.40 in either receptor.

It is worth noting that residue 6.40 is also a bulky residue (Isoleucine) in β_2 AR like Methionine in rhodopsin, and amino acids with bulky, hydrophobic side chains can participate in stabilizing interhelical interactions. In contrast, D2R and D3R have a valine at this position, which is smaller and may be less structurally impactful. This difference in residue identity and physicochemical properties could explain why 6.40 does not appear as a critical connector contact in our analysis of D2R or D3R.

Added in the Sub-section: “The transition between conformational states in β_2 AR as seen via a change in residue community connected to enabler C327(7.54) _ Y326(7.53) and effector R131(3.50)_E268(6.30) contacts”:

Notably, the connector contact between residues I278(6.40) and Y326(7.53), initially broken during the first transition, is reformed in this state, reiterating its structural role in both transitions. The importance of position 6.40 is further emphasized by the fact that it has been reported to stabilize the inactive state of rhodopsin. (Smith et al 2023 <https://doi.org/10.1146/annurev-biophys-083122-094909>)

Also, in Sub-Section “Do the transition enabler and effector residues differ among Dopamine Receptor D2R and D3R subtypes?”:

Five key contacts (TRACs) arise that span the intracellular and transmembrane regions of the receptor. The D1313.49_T682.38 contact at the intracellular region is determined to be a key contact in both D2R and D3R. In the transmembrane region, D80(2.50), S121(3.39) and N422(7.49) residues form two key contacts in D2R that show a hydrogen bond. **Most notably M6.36_Y7.53 contact appears as a connector in D2R which is analogous to the connector contact M6.40_Y7.53 in β_2 AR. This overlap highlights the conserved nature of connector contacts across different GPCRs.**

3. lines 392-394: “We found that in the MD simulations, the deactivation of D2R is faster in time scale by an order of magnitude when compared to β_2 AR and D3R.

This is a remarkable observation. Do the authors have a hypothesis for this difference?

A comprehensive review highlighted that D2R undergoes rapid agonist-induced endocytosis and resensitization, (Cho et al 2010 <https://pubmed.ncbi.nlm.nih.gov/20160122/>; Ji et al 2009 <https://doi.org/10.1073/pnas.0904289106>) whereas D3R exhibits minimal endocytosis and relies more on desensitization mechanisms. This suggests that D2R deactivates more swiftly than D3R. This could be because D2R is involved in motor control and rapid response modulation, requiring fast on/off signaling, while D3R is involved in modulating emotional and motivational states, where prolonged signaling may be beneficial. (Kim 2023 <https://doi.org/10.3390/ijms24076742>)

In addition to the biological/physiological explanation above, there also exists a combinatorial/probabilistic reasoning rationale. The deactivation mechanisms for D2R and D3R exhibit one main difference as determined by DRUMBEAT. D2R deactivation occurs via changes in two TRACs (two events), both occurring at the intracellular regions of the receptor, while D3R deactivation is through three events occurring at the extracellular, connector, and intracellular regions, respectively. The fact that D3R requires more events to occur for deactivation suggests that it would take more time for it to deactivate. We revised the manuscript, as shown below, in accordance to this commentary.

To verify the predicted differences in the transition times for D2R and D3R, we generated chimeras in which the key TRAC residues identified by DRUMBEAT were exchanged between the two receptors. Specifically, residues differing between D2R and D3R TRACs, including F172/L174 in ECL2, position 5.66, and 34.56 in ICL2, were swapped such that each receptor harbored the other's allosteric determinants. We then performed 25 independent molecular dynamics deactivation simulations for each mutant chimera.

The results confirmed the model's predictions: the chimeric D_{2/3}R mutant exhibited a significantly increased deactivation time (median 220 ns versus 70 ns in wild-type D2R), whereas the D_{3/2}R mutant showed a faster deactivation (median 180 ns versus 330 ns in wild-type D3R). These findings demonstrate that the swapped TRACs are indeed critical determinants of deactivation dynamics and directly influence receptor conformational transitions in a manner predicted by our approach.

In light of these new findings, we added the section shown below, along with Table 1, Figure 7, and Table 2.

Added in sub-section “Do the transition enabler and effector residues differ among Dopamine Receptor D₂R and D₃R subtypes?”:

We found that in the MD simulations, the deactivation of D₂R is faster in time scale by an order of magnitude when compared to β_2 AR and D₃R. This could be due to D₂R being involved in motor control and rapid response modulation, requiring fast on/off signaling. In contrast, [Cho et al 2010 <https://pubmed.ncbi.nlm.nih.gov/20160122/>; Ji et al 2009 <https://doi.org/10.1073/pnas.0904289106>) D₃R is mainly involved in emotional and motivational states where prolonged signaling may be beneficial. (Kim 2023 <https://doi.org/10.3390/ijms24076742>) In addition to the biological/physiological explanation, we found that D₂R undergoes deactivation via two key events happening in the receptor whereas D₃R requires three events. This provides a probabilistic/combinatorial explanation for why D₃R takes longer to deactivate.

We also added a new subsection after the section titled “Do the transition enabler and effector residues differ among Dopamine Receptor D₂R and D₃R subtypes?”

Swapping Time-Resolved Allosteric Contacts Confirms the Predicted Determinants of Receptor Deactivation Dynamics

To evaluate whether the TRACs we identified using TRBNM for D₂R and D₃R represent critical determinants of the deactivation transition and whether altering them can modify the transition patterns, we performed reciprocal TRAC-swapping mutations between the two receptors at positions where the TRACs differ (Fig. 7A). Specifically, we mutated residues F172/L174 in ECL2 (TRAC F4.62_F172 in D₃R), position 5.66 (TRAC R5.66_Y5.62 in D₂R) by introducing an R→K mutation in D₂R and a K→R mutation in D₃R, and position 34.56 in ICL2 (TRAC G34.56_S4.38 in D₃R) by inserting a residue in D₂R and deleting it in D₃R (Fig. 7B, Table 1). We conducted 25 deactivation MD simulations for each chimera, obtaining 19 successful deactivation trajectories for the D_{2/3}R mutant and 14 for the D_{3/2}R mutant. In both cases, the time required for deactivation differed significantly from their respective wild-type receptors (Fig. 7C). The average deactivation time for the D_{2/3}R chimera increased from 70 ns to 220 ns, whereas for the D_{3/2}R chimera, it decreased from 330 ns to 180 ns.

Table 1. Swapping mutation between D₂R and D₃R.

TRAC	Mutation in D ₃ R	Mutation in D ₂ R
F4.62_F172 (D ₃ R)	F172L	L174F
G34.56_S4.38 (D ₃ R)	G141 deletion	G144 insertion
R5.66_Y5.62 (D ₂ R)	K216R	R217K

Figure 7. (A) Alignment of residues in D₂ and D₃R TRACs as identified by DRUMBEAT. Red arrows show the positions where mutagenesis was done. (B) Locations of mutations in the receptor. (C) Data for deactivation times for wild-type and mutant D₂R/D₃R.

To quantitatively assess these differences, we performed Mann-Whitney U tests (Table 2). The results confirmed significant differences between wild-type receptors and their respective mutants, with U statistics of 2 ($p < 0.0001$) for D₂R vs D_{2/3}R and 41 ($p = 0.0283$) for D_{3/2}R vs D₃R. Interestingly, not only did the mutations alter the deactivation time scales, but the mutants also behaved in a manner opposite to their respective wild-type receptors. Specifically, the D_{2/3}R chimera exhibited a deactivation time distribution more similar to D₃R than to D₂R ($U = 65.5$ vs D₃R and $U = 2$ vs D₂R), whereas the D_{3/2}R chimera behaved closer to D₂R than D₃R ($U = 30.5$ vs D₂R and $U = 41$ vs D₃R). These findings indicate that the correct combination of TRACs is

required to ensure efficient progression through the deactivation pathway and that swapping TRACs reverses the receptor's deactivation characteristics.

Table 2. Summary of computed statistics for swapping mutation study.

Comparison (vs groups)		Mann-Whitney Test	
		U statistics	p-value
D ₂ R	D _{3/2} R	30.5	0.0114
D ₂ R	D _{2/3} R	2	0
D ₂ R	D ₃ R	9	0.0005
D _{3/2} R	D _{2/3} R	103	0.2793
D _{3/2} R	D ₃ R	41	0.0283
D _{2/3} R	D ₃ R	65.5	0.0498

4. line 399: “The D131(3.49)-T68(2.38) contact at the intracellular region is determined to be a key contact in both D₂R and D₃R”

Is this contact also important in β_2 AR? This would be expected given the role of D3.49 in the salt bridge of the DRY motif.

Thank you for the insightful comment. In response, we examined the D130(3.49)–T68(2.39) contact in β_2 AR to assess whether it plays a role similar to that observed in D₂R and D₃R. While this contact is present in the universal contact graph, it exhibits a low weighted degree that remains low throughout the trajectory, indicating weak dependency with other contacts and limited influence in the broader network. Consequently, it does not appear in the transition-relevant contacts identified by DRUMBEAT. Analysis of the raw trajectory data shows high contact fluctuation throughout the simulation, with no consistent pattern across the conformational transition. These findings suggest that in β_2 AR, this contact is not a stable or functionally significant feature during deactivation. We agree with the reviewer on the critical role of D3.49 (especially in D₂R/D₃R via contact with T2.38), but in β_2 AR, its function appears to be primarily mediated through its interaction with R3.50 within the DRY motif and, to a lesser extent, with E6.30 in the formation of the ionic lock.

5. lines 400-402: “In the transmembrane region, D80(2.50), S121(3.39) and N422(7.49) residues form two key contacts in D₂R that show a hydrogen bond.”

Are these contact also important in β_2 AR? This comparison would be valuable, as these contacts involve highly conserved residues.

We investigated whether the hydrogen bond contacts between D2.50–S3.39 and N7.49–D2.50, highlighted in D₂R, also play a role in β_2 AR.

Our analysis confirms that both contacts are present in β_2 AR and emerge in the DRUMBEAT results. However, they were not in the top of the list of TRACs in β_2 AR, as the major transition signals in this receptor were dominated by other TRACS such as the R3.50–E6.30 ionic lock and C7.54_F8.50. In response to the reviewer's observation, we conducted a more detailed comparison of the inter-residue distances between these residues in the MD simulation trajectories.

Notably, we found that the hydrogen bond network (S3.39–D2.50–N7.49) in β_2 AR also helps stabilize the active state and is consistently disrupted during deactivation, a pattern that closely mirrors what is observed in D₂R. Moreover, the sequence in which these hydrogen bonds are lost upon deactivation is largely preserved between β_2 AR and dopamine receptor D₂R : the D2.50–S3.39 hydrogen bond is the first to break, followed by N7.49–D2.50. Importantly, these residues (S3.39, D2.50, and N7.49) are known to be a part of the conserved sodium-binding site in Class A GPCRs, which plays a key role in modulating receptor activation and stabilization of the inactive state by binding the sodium ion (Katritch et al 2014 <https://pubmed.ncbi.nlm.nih.gov/24767681/>). The disruption of this hydrogen-bonded network prior to deactivation provides mechanistic insight that links this well-established structural motif to dynamic switching between active and inactive states.

This similarity in the role and dynamics of these contacts underscores their conserved functional significance. In light of the value of this new insight, we have added text to the manuscript to highlight this finding along with a Figure S5 added to the Supporting Information. We greatly appreciate the reviewer for drawing attention to this comparative aspect.

Added Lines in sub-section “Do the transition enabler and effector residues differ among Dopamine Receptor D₂R and D₃R subtypes?”:

In D₂R, initially the contacts between D80(2.50), S121(3.39), and N422(7.49) residues stabilize the active state via hydrogen bonding. **Notably, this hydrogen bond network is also present in β_2 AR, consistent with the high conservation of these residues across class A GPCRs. In β_2 AR, the network appears to stabilize the active state and is disrupted as the receptor transitions to the inactive state, paralleling observations in D₂R. Importantly, the sequence in which the hydrogen bonds break is largely preserved in both receptors: the D2.50–S3.39 hydrogen bond is lost first, followed by disruption of the N7.49–D2.50 bond (See Figure S5). Moreover, these residues (S3.39, D2.50, and N7.49) are known to be a part of the conserved sodium-binding site in Class A GPCRs, which plays a key role in modulating receptor activation and stabilization of the inactive state by binding the sodium ion (Katritch et al 2014 <https://pubmed.ncbi.nlm.nih.gov/24767681/>). The disruption of this hydrogen-bonded network prior to deactivation provides mechanistic insight that links this well-established structural motif to dynamic switching between active and inactive states.**

Added SI Figure:

Figure S5 Time-resolved weighted degree plots for D2.50_S3.39 and N7.49_D2.50 contacts from β_2 AR DRUMBEAT results. Similar to what was found for D₂R, these contacts representing a hydrogen bonded core undergo breaking up before deactivation. Notably, the order by which these break is the same as in D₂R, D2.50_S3.39 breaks first followed by N7.49_D2.50

6. lines 405-407: “This contact is unique to the D3R and has been shown via mutation studies to play a significant role in ligand binding although it is not located directly in the ligand binding site.”

Do the authors have a hypothesis for this allosteric effect?

We thank the reviewer for this insightful question. The contact in question, involving F172 in D₃R, is indeed not located directly within the orthosteric ligand binding pocket but has been shown through mutation F172A to abolish antagonist binding (Lundstrom et al 1998 <https://pubmed.ncbi.nlm.nih.gov/9651882/>).

Although F172 is positioned outside the orthosteric ligand-binding site, we hypothesize that it plays a key allosteric role by stabilizing the global architecture of the extracellular region, particularly the arrangement of transmembrane helices and extracellular loop 2 (ECL2), which are known to influence the shape and accessibility of the binding pocket. The functional

relevance of F172 is further supported by its inclusion in a TRAC identified by DRUMBEAT, suggesting that although it does not directly contact the ligand, it participates in a broader network of interactions that regulate conformational transitions and, by extension, receptor functionality.

We have added a sentence to the manuscript in the subsection titled “Do the transition enabler and effector residues differ among Dopamine Receptor D₂R and D₃R subtypes?” to clarify this hypothesis and acknowledge the potential structural role of F172 in modulating ligand affinity via allosteric mechanisms, as follows.

Specifically, the contact F170^{4.62}_F172^{ECL2} at the top of TM4 and in the ECL2 stands out in the D₃R system. This contact is unique to the D₃R and has been shown via mutation studies to play a significant role in ligand binding although it is not located directly in the ligand binding site. We hypothesize that it stabilizes the global architecture of the extracellular region via extracellular loop 2 which can influence the shape and accessibility of the binding pocket. The protein-spanning nature of TRACs in D₃R alludes to the allosteric nature of the receptor’s activity.

7. lines 417-419: “This most likely corresponds to the outward movement of TM7 that we observed in the β_2 AR study and is a preliminary step for deactivation.”

Can this be directly verified in the simulations?

We thank the reviewer for this insightful question. To evaluate whether the observed TRAC (N7.45_L7.41) corresponds to outward movement of TM7, we analyzed the temporal relationship between the TRAC and the breaking of the Y-Y motif (between Y209 on TM5 and Y426 on TM7), a structural marker associated with movement in this region. Specifically, we extracted the frame at which the TRAC intensity of N_L peaks and compared it to the time point of Y-Y motif breaks across individual trajectories.

D ₂ R Trajectory	Time snapshot for Y-Y motif breakage [ns]	N7.45_L7.41 TRAC peak [ns]
1	118	121
2	1 kcal/mol·Å ² restraints step of equilibration	78.9
3	117	1.80
4	120	12.7
5	381	234
6	1 kcal/mol·Å ² restraints step of equilibration	1.20
7	1 kcal/mol·Å ² restraints step of equilibration	9.00

8	1 kcal/mol·Å ² restraints step of equilibration	44.6
9	1 kcal/mol·Å ² restraints step of equilibration	15.7
10	14.5	0.00
11	1 kcal/mol·Å ² restraints step of equilibration	0.10

Our analysis shows that in a subset of trajectories (shown in green in the table above), the N7.45_L7.41 TRAC peaks prior to the Y-Y motif breaking, suggesting it may facilitate the associated helical movement. However, in other trajectories, the Y-Y motif is already disrupted at the start, making it difficult to infer a causal relationship. Thus, while the N7.45_L7.41 TRAC appears consistent with TM7 movement in select cases, this interpretation is not uniformly supported across the dataset.

We agree that our original phrasing was overly definitive and have revised the sentence to reflect this nuance, indicating that the correspondence between N7.45_L7.41 and TM7 movement is suggestive but not confirmed.

The sentence has been changed to in the section “Do the transition enabler and effector residues differ among Dopamine Receptor D₂R and D₃R subtypes?”:

This TRAC may correspond to the outward movement of TM7 observed in the β₂AR study, representing a potential preliminary step in deactivation, though this relationship was not uniformly observed across D₂R trajectories.

8. Figure 4B: The authors show G protein selectivity of different receptor subtypes, but I don't see any discussion in the manuscript. It is tempting to speculate that differences in the deactivation/activation mechanism may be related to G protein engagement and/or selectivity. Perhaps the authors want to explore this point.

We agree with the reviewer on this excellent point --- this is something that we are currently working on and is beyond the scope of this manuscript.

9. line 141: “The DRUMBEAT Method: The details of DRUMBEAT are given in the Supporting Information.”

I did not find the details of the algorithm in the Supporting Information.

We changed this sentence to instead point the reader to the methods section for details, as well as the GitHub, which has more information and a demo for the code.

The sentence in the subsection: “Dynamically Resolved Universal Model for BayEsiAn network Tracking (DRUMBEAT) applied to analyze temporal transition events in proteins” now reads:

The details of DRUMBEAT methodology can be found in the Methods section as well as the Github (github.com/bandyt-group/drumbeat) which also provides a demo for the code.

Minor Revisions

1. The abstract emphasizes drug design, but the manuscript focuses on protein dynamics. Drug design represents a potential application not directly explored in this work.

We thank the reviewer for the comment. It is true that this work focuses on characterizing the transitions observed in protein dynamics, with the end goal of finding small molecule binding sites that are specific to target proteins nestled in the residues that enable transitions. We mentioned drug design because this is our end goal, and to show a therapeutic strategy emerging from this method. This is why we emphasize drug design in the abstract since it is arguably the most impactful outcome of the DRUMBEAT methodology.

2. lines 338-339: “For instance, the salt bridge between R131(3.50) and E268(6.30) is one of the established indicators of rhodopsin deactivation.”

Consider referencing Ballesteros et al. (JBC, 2001; doi: 10.1074/jbc.m103747200) for early ionic lock studies. Also note that position 6.30 varies across class A GPCRs; the R3.50-E6.30 salt bridge exists primarily in specific subfamilies (mainly amine receptors and opsins).

We thank the reviewer for the comment. We have added the reference to the manuscript and the note as mentioned by the reviewer as follows:

Added to subsection titled “The transition between conformational states in β_2 AR as seen via a change in residue community connected to enabler C3277.54 _ Y3267.53 and effector R1313.50 _ E2686.30 contacts”:

For instance, the salt bridge between R131^{3.50} and E268^{6.30} is one of the established indicators of rhodopsin deactivation, and R131^{3.50} is part of conserved ED/R_Y motif on TM3 across all class A GPCRs. It is worth noting that the position 6.30 has variation across different class A GPCRs, leading to the R3.50_E6.30 salt bridge primarily existing in amine receptors and opsins. [Ballesteros et al 2001 <https://pubmed.ncbi.nlm.nih.gov/11375997/>]

3. The authors could consider discussing how this approach compares to other recent methods for analyzing conformational transitions in GPCRs, such as Markov state models or deep learning approaches.

We thank the reviewer for the comment.

Importantly, we have deliberately chosen naturally interpretable ML models over various DL approaches. While the latter are undoubtedly promising in the context of MD simulations’

generation, sampling, and extension (D'Hondt et al. 2025 <https://doi.org/10.1186/s13321-025-00985-7>, Lewis et al 2025 <https://www.science.org/doi/10.1126/science.adv9817>), their applicability to the MD data analysis is limited to dimensionality reduction/feature extraction (Zhang et al 2025 <https://doi.org/10.1093/bioinformatics/btaf321>, Mustali et al 2023 <https://pubs.rsc.org/en/content/articlelanding/2023/ra/d3ra06375e>) or supervised learning (classification) with at most post-hoc explainability, such as one-way feature importance ranking (Plante et al 2019 <https://doi.org/10.3390/MOLECULES24112097>, Plante et al 2021 <https://doi.org/10.3390/MOLECULES26103059>, Do et al 2022 <https://doi.org/10.1021/acs.jctc.1c01055>, Moraes doe Santos et al 2024 <https://doi.org/10.1038/s41598-024-72842-w>). This is in sharp contrast with our BN approach --- although it is as fully data-driven and as sensitive to rare events and nonlinearity as DL, its deliverables are not the classification accuracy and/or variable importance rankings, but rather the intrinsically interpretable multi-way interactions represented as dynamic network structures (directed acyclic graphs, DAGs). Concerning MSM, while it remains a powerful method for analyzing protein dynamics, there are several aspects in which DRUMBEAT differs from MSM:

1. MSM discretizes the protein configurational space into metastable states and then models the protein dynamics as transitions between these states. A major distinction of MSM is the imposition of a Markovian relationship among the states. In contrast, DRUMBEAT employs BNs and hence can model more complex, non-stationary co-dependencies beyond what MSM offers.
2. One major challenge in MSM is the proper choice of collective variables (CVs) to define the underlying states. This becomes increasingly difficult for larger systems where a limited number of human-interpretable CVs may not be able to model the dynamics in a way that retains Markovian character. This limitation is addressed by applying ML methods such as VAMP-Net, but then the resulting CVs are no longer interpretable (Konovalov et al 2021 <https://pubmed.ncbi.nlm.nih.gov/34604842/>). In contrast, DRUMBEAT uses structure-wide protein features such as inter-residue contacts or interaction energy to model the system in an unbiased manner, without relying on pre-defined CVs. This facilitates DRUMBEAT to be applied to large complex systems, without sacrificing interpretability.
3. DRUMBEAT can identify protein features critical for dynamic transitions and thus can assist in constructing CVs for MSMs.
4. DRUMBEAT gives insights into allosteric dependencies among protein features that enable transitions. These insights are less straightforward to obtain from MSMs, which focus on dynamic relationships across whole protein states, rather than individual structural features.

Reviewer #4 (Remarks to the Author)

General Comments

This work proposes the ML approach for obtaining and analyzing GPCRs conformational transitions.

The ML model is built in relation to allosteric regulation of GPCRs activity, aiming at “pinpointing residue communities regulating conformational state ensembles”. The latter is another way of describing allosteric communication and signalling via dynamics of the whole protein (PMID: 26562539, PMID: 30439587, PMID: 26939022).

1. The most important aspect of allosteric communication – causality, which should be discussed explaining the link between the structural dynamics and its reflection in proposed here DRUMBEAT method. The causality of signalling can be checked with the help of allosteric signalling and probing maps precalculated for different GPCRs (<https://allomaps.bii.a-star.edu.sg>, PMID: 32392302) – but this is fully to authors’ discretion.

We thank the reviewer for highlighting the importance of causality in allosteric communication and for suggesting comparative analysis with AlloMAPS. Our primary aim in this work is to identify and characterize the residue contacts and events that precede and accompany conformational transitions, specifically those that enable receptor deactivation. While methods such as AlloMAPS are powerful for mapping general information transfer pathways and steady-state allosteric signaling, DRUMBEAT is uniquely focused on revealing the temporally ordered molecular events associated with specific functional transitions, rather than inferring the direction or pathway of information transfer per se.

Given this distinction in the scope and mechanistic targets of our approach, we believe a direct comparison with AlloMAPS is outside the intent of our current study. We agree, however, that integrating such information-theoretic causality analyses represents an important direction for future research, and we are actively developing methods to infer allosteric signaling pathways and information transfer—work that will be the subject of a forthcoming manuscript.

2. It is questionable to understand how well few top-ranking contacts reflect large conformational changes taking place upon receptor’s functional regulation. What other network characteristics reflecting above changes authors may think can be used? It is relevant to Enabler-Effector and Connector Contacts, which are characterised by subnetworks – these are seemingly elements that may answer above questions. Is it possible to obtain their quantitative description?

We thank the reviewer for the thoughtful comment.

We have examined how the DRUMBEAT predicted TRACS affect the conformational changes observed in the structural dynamics. Figure 3D for β_2 AR and Figures 5D and 5F for D₂R and

D₃R, respectively, show the time-related events of how TRACS temporally propagate the transition spatially through the receptor structures.

With regards to how well the top ranking TRACs reflect large conformational changes: Some of the top-ranking TRACS or residue communities that enable and effect transition indeed reflect large conformational change themselves or lead to conformational changes in other residue pairs. We added Table S1 to the Supporting Information (See below) which shows the TRACS residue pairs and their functional relevance and impact on microswitch residues that show a conformational change upon activation or deactivation.

With respect to other network properties, in this study, we primarily focus on weighted degree as a network property for characterizing key contacts, due to its robustness and interpretability, as demonstrated in our previous work. (Mukhaleva et al 2024 [https://www.jbc.org/article/S0021-9258\(24\)01863-5/fulltext](https://www.jbc.org/article/S0021-9258(24)01863-5/fulltext)) However, we fully agree that other network-based metrics could offer additional insights into the broader conformational changes associated with receptor function.

In recent work, we have further explored the use of network entropy as a dynamic, graph-level (as opposed to node- or edge-level local context) measure and demonstrated its ability to capture folding and unfolding transitions in the Fip35 protein system. While these results are not included in the present manuscript, they are part of a forthcoming publication that further advances the DRUMBEAT methodology and its applications to protein dynamics.

Regarding the reviewer's suggestion to provide a quantitative description of the Enabler, Effector, and Connector subnetworks: this is indeed possible. For instance, one could compute metrics such as the average weighted degree, clustering coefficient, or intra-subnetwork edge density to characterize the internal connectivity of each TRAC. Additionally, measuring the relative change in these metrics across conformational states may provide insight into their roles during transitions. We recognize the value of this suggestion and will aim to incorporate such quantitative descriptors in our future work.

Table S1: Summary of TRACS determined by DRUMBEAT method for each receptor. A brief description of the functional relevance for each TRAC is also provided. Each TRAC is determined to be either a known microswitch or not, highlighting DRUMBEAT's ability to recapitulate known microswitches while also providing insight on new contacts that facilitate transition.

Protein	DRUMBEAT TRAC residue contacts	Receptor Function Relevance	Known Microswitch
β2AR	R3.50_E6.30	Ionic lock; stabilizes inactive state in many class A GPCRs.[1]	Yes
	C7.54_F8.50	Part of NPxxY/F motif; essential for G-protein coupling.[2]	Yes
	C7.54_V1.53	TM1–TM7 contact; aids structural stability during transitions.	No

	R8.51_C7.54	Supports helix 8-TM7 interface; involved in signaling specificity.	No
	F8.50_Y7.53	Aromatic cluster; key in conformational switch (NPxxY region).[3]	Yes
	S8.47_Y7.53	Modifies NPxxY switch dynamics; stabilizes helix 8 rearrangement.	No
	L3.43_Y7.53	Facilitates TM3-TM7 cross-talk during receptor transitions.	No
D ₂ R	D2.50_S3.39	Core sodium pocket; allosteric regulation in class A GPCRs.[4]	Yes
	N7.49_D2.50	Links NPxxY and sodium pocket; disrupted upon deactivation.[4]	Yes
	N7.45_L7.41	TM7 stability; may support NPxxY function (not a classic switch).	No
	D3.49_T2.38	DRY motif-TM2 link; stabilizes inactive state (subtype specific).	No
	R5.66_Y5.62	TM5 interaction; may aid in cytoplasmic region stability.	No
D ₃ R	D3.49_T2.38	DRY motif-TM2 link; stabilizes inactive state (subtype specific).	No
	C6.47_F6.51	TM6 toggle switch region; central in activation.[5]	Yes
	M3.56_Y3.51	TM3 contacts; support DRY motif packing and transitions.	No
	F4.62_F172	TM4/loop aromatic contact; may influence subtype signaling.	No
	G34.56_S4.38	TM4 local contact; role not classically recognized.	No

(Table references to be added to revised manuscript according to journal formatting: 1. Latorraca et al 2017 <https://pubs.acs.org/doi/10.1021/acs.chemrev.6b00177>; 2. Fritze et al 2003 <https://www.pnas.org/doi/full/10.1073/pnas.0435715100>; 3. Zhou et al 2019 <https://doi.org/10.7554/eLife.50279>; 4. Katritch et al 2014 <https://doi.org/10.1016/j.tibs.2014.03.002>; 5. Shi et al 2002 <https://doi.org/10.1074/jbc.M206801200>)

- The conservatism of residues involved into function related conformational transitions is a very interesting and important aspect, as well as diversity observed in different subtypes of receptors. It would be interesting to see conservation and variability of signalling's causality, in which these residues are involved.

We thank the reviewer for the comment. We looked into this question in the context of the importance of multiple enabler and effector residue interactions, or TRACS identified in this study, being common across β_2 AR and the two dopamine receptors, D₂R and D₃R. For example, we analyzed the role of the conserved residue contacts between D2.50, S3.39 and N7.49 that resulted in being common to D2R and β_2 AR studied here. Interestingly, the order of events by which the hydrogen-bonded core breaks is also similar for D2R (D_S breaking -> N_D breaking) and β_2 AR. This provides evidence that although there are notably different mechanisms by

which deactivation can happen in different GPCRs, the strong conservation between receptors leads to similarities in the order of signaling that may occur before the deactivation proceeds.

We revised the manuscript as follows:

Added Lines in sub-section “Do the transition enabler and effector residues differ among Dopamine Receptor D₂R and D₃R subtypes?”:

In D₂R, initially the contacts between D80(2.50), S121(3.39), and N422(7.49) residues stabilize the active state via hydrogen bonding. Notably, this hydrogen bond network is also present in β_2 AR, consistent with the high conservation of these residues across class A GPCRs. In β_2 AR, the network appears to stabilize the active state and is disrupted as the receptor transitions to the inactive state, paralleling observations in D₂R. Importantly, the sequence in which the hydrogen bonds break is largely preserved in both receptors: the D2.50–S3.39 hydrogen bond is lost first, followed by disruption of the N7.49–D2.50 bond (See Figure S5). Moreover, these residues (S3.39, D2.50, and N7.49) are known to be a part of the conserved sodium-binding site in Class A GPCRs, which plays a key role in modulating receptor activation and stabilization of the inactive state by binding the sodium ion (Katritch et al 2014 <https://pubmed.ncbi.nlm.nih.gov/24767681/>). The disruption of this hydrogen-bonded network prior to deactivation provides mechanistic insight that links this well-established structural motif to dynamic switching between active and inactive states.

Added SI Figure:

Figure S5 Time-resolved weighted degree plots for D2.50_S3.39 and N7.49_D2.50 contacts from β_2 AR DRUMBEAT results. Similar to what was found for D₂R, these contacts representing a hydrogen bonded core undergo breaking up before deactivation. Notably, the order by which these break is the same as in D₂R, D2.50_S3.39 breaks first followed by N7.49_D2.50

4. The method should be applied to additional receptors (not only Dopamine family) to show that it is generic. What would be the “toolset” of parameters for anticipated generalized analysis. Please, discuss it.

We appreciate the reviewer’s comment. The DRUMBEAT method is inherently system-agnostic and does not require tailoring a new model for each protein. In this study, β_2 AR was used as a structural prototype to illustrate the approach, and D₂R/D₃R were chosen as high-value drug targets where subtype-specific regulatory mechanisms remain poorly understood and are relevant for allosteric modulator discovery. While the resulting insights—such as the identity of TRACs and specific transition mechanisms—are system-specific, this intrinsic generalizability is a strength of the method rather than a limitation.

The key parameters that can be adjusted for any application include:

1. Mutual information (MI) cutoff – controls the number of residue contacts incorporated into the universal graph.
2. Trajectory sampling scheme – allows targeted enrichment of specific trajectory regions (e.g., near the transition state region for β_2 AR) when building the universal graph.
3. Scanning resolution – smaller temporal windows yield higher-resolution TRAC analysis suitable for fast transitions; larger windows give a coarser but faster overview for initial screening.
4. TRAC selection criteria – e.g., focusing on the top N TRACs ranked by weighted degree.

Details of these parameters and the full DRUMBEAT protocol are provided in our public GitHub repository (www.github.com/bandyt-group/drumbeat).

Ultimately, these parameters are adjusted according to the study's goals (e.g., dissecting the transitions between macrostates versus microstates) and computational resources rather than to the protein system in question. The framework is universally applicable to any protein system, including GPCRs and beyond (e.g., peptide folding, nanobodies), with the same underlying methodology. In this sense, the generalizability of DRUMBEAT is intrinsic and orthogonal to the protein family under investigation.

We have added the following text to the discussion to address the comment made by the reviewer:

Added text to Discussion:

The implications of this work stem from both the innovative methodology and the results it produced. The DRUMBEAT approach introduces a novel way to study protein dynamics beyond their stationary states. By using a universal graph to establish probabilistic dependencies across conformational states and a scanning technique for temporal resolution, we identified key residues and events driving transitions. This dual capability provides critical insight into the most challenging regions of dynamic space—the transitions—by pinpointing when and where the protein deviates from equilibrium. Furthermore, the results propose new, collective reaction coordinates for transitions that move beyond traditional single-coordinate models (e.g., distances between two residues) to incorporate coordinated motions among multiple residues. This opens avenues for advanced sampling techniques to model transitions more efficiently. **Importantly, DRUMBEAT is a protein system-agnostic framework whose core parameters, such as the contact inclusion cutoff, trajectory sampling scheme, temporal scanning resolution, and TRAC selection criteria, can be tuned to accommodate different study goals and available computational resources without altering the underlying model. This flexibility enables its application well beyond dopamine receptors, including other GPCR classes and non-GPCR proteins such as nanobodies or folding peptides, with the same procedural framework. The ability to adapt**

DRUMBEAT parameters allows researchers to balance resolution and computational cost while preserving the method's capacity to resolve the temporally ordered events that shape conformational transitions. In addition to generalizability, the reproducibility of results across different GPCRs and the identification of conserved residues within evolutionary families underscore the methodology's robustness. These findings not only shed light on the mechanics of GPCR transitions but also highlight residues critical for receptor family function, paving the way for experimental validation, mutation studies, and therapeutic targeting.

Reply to Reviewer #1:

Reviewer comment shown in verbatim in *italics*. Response follows in normal font with revisions where applicable.

- 1) *The commonly accepted definition of microswitches describes them as residues triggering conformational changes leading to the active-state receptor structure. In contrast, Authors would like to present ‘inverted microswitches’ leading to the inactive-state receptor structure. In this sense detection of such ‘transition enablers’ or ‘effector communities’, as they call them, in MD trajectories by machine learning is worth considering by Nature Communications. Nevertheless, I still have doubts regarding the terms ‘enablers’ and ‘residue communities’.*
- What is more, if these ‘inverted microswitches’ do not overlap with typical microswitches (Table S1) it means that the receptor activation pathway observed as series of conformational changes of the receptor core is intrinsically different comparing the receptor deactivation pathway. Authors should discuss it and add references describing such receptor behavior among various GPCR classes and hypothesize a possible evolutionary explanation. The newly formed inactive state should be reversible. If it is formed in a different pathway, is it indeed the same to the atomic details as before?*

The term “microswitch” in the context of G protein-coupled receptors (GPCRs) literature emerged from comparative analysis of high-resolution structures of the agonist-bound active intermediate state and inactive state of class A GPCRs (<https://doi.org/10.1146/annurev-pharmtox-032112-135923>; <https://doi.org/10.7554/eLife.50279>). These studies listed the residues/residue pairs that show local conformational changes in the static structures of the active state in comparison to the inactive state as *microswitches*. Our study identifies residue pairs that enable conformational changes but do not necessarily show conformational changes after activation or deactivation. Microswitch residues are not necessarily transition enablers. Regarding the question of the pathways of deactivation being the same as activation pathways: In the case of β_2 AR, the macrostate transitions that we observe in nearly all the trajectories (TM7 moving away from TM5 followed by TM6 moving towards TM3) are evidence that the reverse order of these events corresponds to the activation pathway (<https://doi.org/10.1038/nature19107>). Having said that, we do not want to claim that this is the case for all receptors, but rather is a special case for β_2 AR. The DRUMBEAT methodology merely identifies the enabler residues of deactivation and does not directly imply that they are the same as those involved in activation pathways. We have explained these details in the revised manuscript.

From the **Discussion** section (added the following text in red):

This work addresses a critical challenge in protein dynamics: identifying key regions and residues, or "allosteric residue communities", that facilitate transitions between conformational states and ascertaining how these communities behave temporally. The DRUMBEAT methodology developed in this study uncovers the temporal dynamics and mechanistic underpinnings of conformational transitions, in the β 2AR and two highly homologous and sought after drug targets, namely dopamine D2R and D3R receptors. We identified Time-Resolved Allosteric Communities (TRACs) that correspond to key sub-regions of the protein effecting the conformational changes and how their manifestations coincide with the transition of the receptor from active to inactive or vice versa. **In contrast to 'microswitches', which are defined as residues/residue pairs that show local conformational changes in the static structures of the active state in comparison to the inactive state, this study identifies residue pairs that enable conformational changes but do not necessarily show conformational changes. Although DRUMBEAT may identify microswitches, it also can uncover new residues, or regions of the protein, that are critical for activation, but do not fall in the category of microswitches.** The results emphasize how probabilistic connectivities can identify critical contacts and their allosteric coordination during key transition events, providing a fresh perspective on receptor transition.

In Section titled "The transition between conformational states in β 2AR as seen via a change in residue community connected to enabler C3277.54 _ Y3267.53 and effector R1313.50_E2686.30 contacts"

Figure 3D shows the residue contacts involved in the enabler and effector signals in the three distinct states delineated from PCA. In state 1, the central contacts in the enabler TRAC break and make contacts involving residues C327^{7.54} and Y326^{7.53} as TM7 moves away from TM3. During this change, a large portion of breaking and forming contacts between C327^{7.54} and F332^{8.50} also occurs between TM7 and helix 8. This transition gives rise to state 2, an intermediate state between active and inactive conformations, a finding that follows the report²⁵ from which the trajectories were obtained. The precise structural motion and key residues involved provide a new and deeper understanding of the transitioning to the intermediate state. After some time in this intermediate state, the second transition associated with the effector signal occurs, leading to state 3. This transition occurs via TM6 swinging into the G-protein binding pocket towards TM3, ultimately forming a salt bridge between the central contact of the effector TRAC, R131^{3.50} and E268^{6.30}, indicating the transition of receptor into inactive state.⁴¹ **Studies indicate the reverse sequence corresponds to the activation pathway for β 2AR. (<https://doi.org/10.1038/nature19107>). However, this may be unique to β 2AR and should not be generalized to all receptors. The DRUMBEAT methodology merely identifies enabler residues of deactivation and does not directly imply equivalence with those**

involved in activation pathways. We note here that the residue R3.50 could also form a salt bridge with its neighbor D3.49 as seen in the crystal structures of the inactive state of some class A GPCRs.⁴² We examined this contact in the β_2 AR trajectories and found it does not consistently form upon deactivation, in line with previous reports of significant basal activity for β_2 AR leading to the lack of formation of the R3.50_D3.49 salt bridge in the inactive state.⁴³ Notably, the connector contact between residues I278^{6.40} and Y326^{7.53}, initially broken during the first transition, is reformed in this state, reiterating its structural role in both transitions. The importance of position 6.40 is further emphasized by the fact that it has been reported to stabilize the inactive state of rhodopsin.⁴⁴ Previous studies corroborate the importance of these key residues. For instance, the salt bridge between R131^{3.50} and E268^{6.30} is one of the established indicators of rhodopsin deactivation⁴⁵, and R131^{3.50} is part of conserved ED/R_Y motif on TM3 across all class A GPCRs.⁴⁶ It is worth noting that the position 6.30 has variation across different class A GPCRs, leading to the R3.50_E6.30 salt bridge primarily existing in amine receptors and opsins.⁴⁷ Also, Y326^{7.53} contributes to GPCR activation via the conserved NPxxY motif.⁴⁸

- 2) *And, what is more important, do e.g. 180 and 220 ns structures of D2/3 or D3/2 chimeras indeed represent their inactive state or just a near-inactive transition state?*

We compared the conformations resulting from the deactivation of D2/D3 and D3/D2 chimeras to the conformational ensemble generated using MD simulations starting from their respective inactive state structures. As shown in Fig. S1 and S2, there is a significant overlap between the two ensembles.

- 3) *Fig. 8 – Authors still has not provided any proof about this allosteric site.*

Thank you for the suggestion. We are currently working on identifying small-molecule allosteric modulators that occupy the predicted allosteric site --- to be published in our future works.

- 4) *What is the percentage of ‘failed’ simulations when this time cholesterol was included in the systems? Did it improve?*

We incorporated cholesterol into new MD simulations of chimeric D2R and D3R systems. In the D2/3R chimeric and D3/2R chimeras, 25% and 44%, respectively, of the trajectories failed to deactivate, resulting in an improvement in the number of successful deactivation MD trajectories.